# Cryo-EM snapshots of a native lysate provide structural insights into a metabolon-embedded transacetylase reaction

Christian Tüting [1,7], Fotis L. Kyrilis [1,2,7], Johannes Müller [2], Marija Sorokina[2,3,4], Ioannis Skalidis [1,2], Farzad Hamdi [1], Yashar Sadian[5] & Panagiotis L. Kastritis [1,2,6 ✉]

Found across all kingdoms of life, 2-keto acid dehydrogenase complexes possess prominent metabolic roles and form major regulatory sites. Although their component structures are known, their higher-order organization is highly heterogeneous, not only across species or tissues but also even within a single cell. Here, we report a cryo-EM structure of the fully active *Chaetomium thermophilum* pyruvate dehydrogenase complex (PDHc) core scaffold at 3.85 Å resolution (FSC = 0.143) from native cell extracts. By combining cryo-EM with macromolecular docking and molecular dynamics simulations, we resolve all PDHc core scaffold interfaces and dissect the residing transacetylase reaction. Electrostatics attract the lipoyl domain to the transacetylase active site and stabilize the coenzyme A, while apolar interactions position the lipoate in its binding cleft. Our results have direct implications on the structural determinants of the transacetylase reaction and the role of flexible regions in the context of the overall 10 MDa PDHc metabolon architecture.

[1] Interdisciplinary Research Center HALOmem, Charles Tanford Protein Center, Martin Luther University Halle-Wittenberg, Kurt-Mothes-Straße 3a, Halle/Saale, Germany. [2] Institute of Biochemistry and Biotechnology, Martin Luther University Halle-Wittenberg, Kurt-Mothes-Straße 3, Halle/Saale, Germany. [3] RGCC International GmbH, Baarerstrasse 95, Zug 6300, Switzerland. [4] BioSolutions GmbH Weinbergweg 22, 06120 Halle/Saale, Germany. [5] Bioimaging Center (cryoGEnic), Université de Genève, Sciences II, 1211 Genève 4, Switzerland. [6] Biozentrum, Martin Luther University Halle-Wittenberg, Weinbergweg 22, Halle/Saale, Germany. [7] These authors contributed equally: Christian Tüting, Fotis L. Kyrilis. ✉email: panagiotis.kastritis@bct.uni-halle.de

The 2-keto acid dehydrogenase complexes comprise a conserved family of giant enzymatic assemblies spanning 4–10 megadalton in size. Such massive complexes, dubbed as "metabolons"[1], play an essential role in carbon entry into the tricarboxylic acid cycle (Krebs cycle, TCA)[2–6], in sugar tuning and amino acid degradation[7] and are thus major metabolic checkpoints. Inborn errors can be deleterious, with clinical symptoms including developmental delay[8], subacute necrotizing encephalomyelopathy (also called Leigh syndrome)[9], encephalopathy[10], and microcephaly[11,12]. Recently, this complex family was also found to perform moonlighting functions[13], being involved in oxidative stress and its response[14], and overall, is systematically identified as a key factor in disease phenotypes, including cancer[15] and neurodegeneration[16,17]. Consequently, its structural characterization is essential, preferably from endogenous material, for understanding molecular and structural features important for 2-keto acid dehydrogenase complex function.

Despite their importance, structural data have been limited to the characterization of their isolated components and not their interactions in a native context due to their sheer size, high heterogeneity, flexibility, and stoichiometric variation[3]. The eukaryotic pyruvate dehydrogenase complex (PDHc), as the most studied family member, is composed of multiple copies of the E1, E2, E3BP (also known as protein X), and E3 enzymes, which all work in concert to perform the dedicated pyruvate oxidation reaction (Supplementary Fig. 1). Architecturally, the core consists of 60 subunits that form a near-symmetric icosahedral structure, formed by 20 trimeric subunits at each corner. In fungi, the core-forming subunit is exclusively E2[18], whereas in mammals the core is composed of a mixture of E2 and E3BP in either a 48:12[19] or 40:20 E2:E3BP ratio. E3BP is evolutionary distinct in fungi, localizing inside the E2 core[20]. Both, the E3BP of fungi, due to its localization inside the core[20,21], and the mammalian E3BP, due to a loss-of-function mutation in the active site[22], are catalytically inactive. The flexible N-ter of E2 and E3BP tether E1 and E3, respectively, in relative proximity to the core[21], forming a local reaction chamber, the pyruvate dehydrogenase factory organization[21]. In this reaction chamber, the lipoyl domain (LD) transport reaction intermediates to the respective active sites. The number of LDs varies in each N-ter, from a single copy (e.g., in yeast and C. thermophilum), to two (e.g., in human) or three (e.g., in Escherichia coli).

Mechanistically, in the first reaction step during pyruvate oxidation, pyruvate gets decarboxylated by E1 and the remaining acetyl-moiety is transferred to the lipoate, which is covalently attached at a conserved Lys in the LD. This reaction is the major regulatory checkpoint of the PDHc and is regulated by phosphorylation and dephosphorylation of E1[23]. After the decarboxylation of pyruvate, the acetylated LD undergoes extensive movement to localize at the E2 core, where the transacetylase reaction takes place. In this reaction, the acetyl-moiety is transferred to a Coenzyme A (CoA), which then enters the Krebs cycle. For regeneration, the now reduced lipoate is reoxidized at the active site of E3, where a FAD is covalently bound. Eventually, the reduced $FADH_2$ is recovered by transferring the two protons and two electrons onto an $NAD^+$ molecule. This reaction cycle requires considerable flexibility of the LD which is delivered by the unstructured regions at the N-ter in E2 and E3BP via a dedicated "swinging arm" mechanism[24], but details on the transiently formed LD interfaces remain unknown in the native context.

Considering the complex role of the E3BP and its stoichiometric variation across species[20,25], or even within single species[20], a complete model of the E2/E3BP core scaffold has not been yet resolved. As a PDHc core scaffold, we describe the native assembly of E2 and E3BP which is essential to understand the localization of E1 and E3, respectively, as well as the relative proximity of all proteins and the LD. Only recent cryo-EM studies have allowed the structural characterization of Neurospora crassa PDHc core scaffold where three helical elements were suggested to reside in identified tetrahedral densities[20]. In addition, C. thermophilum endogenous PDHc has been resolved at low resolution[21], a study which confirmed the localization of E3BP in fungi, its copy numbers, and proposed a hypothetical intramolecular E3BP interface[21].

The abovementioned recent advances in the understanding of the PDHc core scaffold were allowed by a combination of cryo-EM with complementary biochemical, biophysical, and structural methods[20,21] but have not yet elucidated the endogenous core scaffold structure of PDHc. Lacking this critical knowledge, architectural details of the PDHc metabolon cannot be systematically derived, which hinders structure–function investigation of the pyruvate oxidation reaction and especially, the essential production of acetyl-CoA through the transacetylase reaction.

Here, we resolve the complete, active, endogenous PDHc core and the complete scaffold at 3.85 and 6.21 Å resolution, respectively (FSC = 0.143), and systematically probe its composition by integrative structural biology. By combining biochemical, biophysical, and biocomputational analysis, we investigate the kinetics, overall architecture, all interfaces present, evolutionary implications of residing common folds, and, ultimately, visualize the transacetylase reaction within the context of the native, active PDHc metabolon.

## Results

**Biochemical analysis elucidates an active PDHc metabolon in a heterogeneous, high-molecular-weight cell extract fraction.** We aimed to characterize the active PDHc metabolon from endogenous material, an important step to derive structure–function relationships for the pyruvate oxidation reaction. To accomplish this, we established fractionation of megadalton (MDa) range complexes from C. thermophilum cell extracts (Fig. 1a) while monitoring PDHc activity. Note that the growth state of C. thermophilum before harvesting significantly impacted the retrieved PDHc activity and was properly adjusted to recover optimal PDHc activity (see Methods). To unambiguously correlate activity to the presence of all PDHc subunits, we performed immunoblotting experiments, i.e., western-blots (WBs), using antibodies specifically designed against each respective recombinant protein assembling PDHc (Fig. 1b; Supplementary Fig. 2; and Supplementary Table 1). These data provide an orthogonal validation and co-identification of co-eluting PDHc subunits analyzed by label-free mass-spectrometry in the same fractions[21].

To retrieve kinetic parameters of the PDHc metabolon, we determined quantitative kinetic constants for pyruvate and CoA. C. thermophilum PDHc actively converts pyruvate to acetyl-CoA, a multistep reaction that requires three distinct active sites (Fig. 1c). The cofactors (a) thiamine diphosphate (TPP), bound by E1, (b) lipoate, bound by the LDs of C. thermophilum E2 and E3BP, specifically Lys75 and Lys78, respectively, and (c) FAD, bound by the E3 subunit, are attached in their respective active sites. We determine the apparent $K_M$ values of the soluble substrates pyruvate and CoA at $148 \pm 23$ and $68 \pm 11$ μM, respectively (Fig. 1d). Values are comparable to those reported for the mammalian PDHc isolated from human cancer cell tissue[26,27]. These results demonstrate that our simple, single-step enrichment procedure to retrieve native PDHc directly from cell lysate is highly efficient, maintaining an assembled native complex of notable activity.

**Resolving the native core at 3.85 Å from a eukaryotic PDHc metabolon.** To structurally characterize native PDHc, we vitrified the fractionated extract and collected 2808 micrographs at a

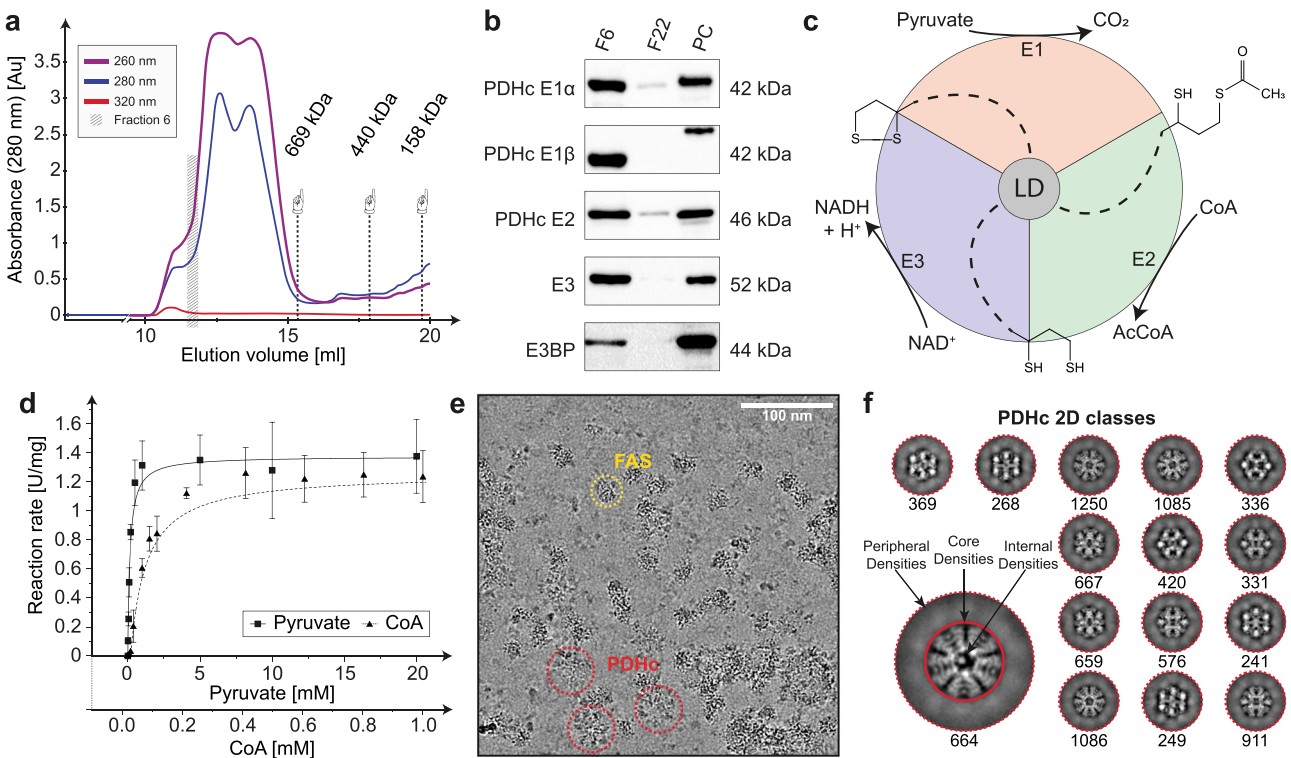

**Fig. 1 Biochemical characterization of the native pyruvate dehydrogenase metabolon. a** Size-exclusion chromatography (SEC) profile of *C. thermophilum* cell extract. Fraction 6, corresponding to ~10 MDa complexes, further analyzed in this study, is highlighted. The retention volume of high-molecular-weight standard proteins are shown and includes thyroglobulin (669 kDa), ferritin (440 kDa), and aldolase (158 kDa). **b** Detection of all components of the PDHc metabolon ($n = 1$). Fraction 6 (F6), Fraction 22 (F22; negative control), and recombinant protein, used for antibody production (PC positive control) were analyzed. The displayed molecular weight corresponds to the recombinant protein. **c** Reaction pathway during pyruvate oxidation. The reaction is mediated by the lipoate (shown as skeletal structural formula) covalently bound to the LD. The reaction requires three distinct active sites at E1, E2, and E3 of PDHc to occur. **d** Enzymatic characterization of native PDHc. Pyruvate and CoA were used in the shown concentration range and the reaction rate was fitted with a hyperbolic function following Michaelis–Menten kinetics. Center of error bars correspond to the mean value, error bars correspond to standard deviation derived from $n = 3$ independent biological samples (technical duplicates in each). **e** Representative micrograph of F6 after denoising. PDHc in higher-order assemblies is visible in a heterogeneous environment, e.g., including fatty acid synthase molecules (FAS). **f** 2D classes of PDHc retrieved from cryo-EM single-particles. External, core, and internal densities are clearly distinguishable. Numbers below representative class averages correspond to the number of particles included in each class.

200 kV Thermo Fisher Glacios cryo-microscope equipped with a Falcon 3 direct electron detector under low dose. Given that higher-order assemblies of notable size exist in this fraction (Fig. 1e), pixel size was kept at 1.57 Å to increase the statistics of observed MDa complexes (see Methods). Our imaging protocol allowed to map an overall cell extract area of 1158.80 µm². Data show that PDHc was already prominently present within those micrographs (Fig. 1e), even in a native fraction where other biomolecules of notable size co-elute. PDHc also often participates in even larger protein communities[21]. Prominent 2D averages were retrieved for PDHc after localizing the metabolon and performing image processing (Fig. 1f) and eventually included 10249 single-particles for further analysis. Densities around the cores were averaged to focus on resolving the core scaffold (Fig. 1f).

The reconstruction of the core with an application of icosahedral symmetry was resolved at 3.85 Å resolution (FSC = 0.143) (Fig. 2a and Supplementary Fig. 3), where the backbone is clearly traceable and side chain resolution is often visible (Fig. 2b). The map's final resolution allowed building a refined model of the *C. thermophilum* PDHc core. The model compares well to the previous lower-resolution counterpart and exhibits a high overall root-mean-square-deviation ($RMSD_{overall} = 1.95$ Å). However, a major improvement in atom placement is derived with the 3.85 Å map as shown by the increased calculated correlation coefficients (*cc*)

between map and model (*cc* = 0.9 ± 0.1 for the new model versus *cc* = 0.4 ± 0.2 for the previous model) (Fig. 2b).

The overall icosahedral core is stabilized by two interfaces with distinct energetic signatures, which were calculated with HADDOCK (Fig. 2c). In particular, the intra-trimeric interface, buries a large surface area of ~5000 Å², while the inter-trimeric interface buries only ~1500 Å². Such differences point to the former sharing similarities to permanent interfaces[28], whereas the latter to interfaces found in transient protein–protein interactions[29]. The size of the interfaces may corroborate the conservation of an assembly mechanism for the core where the E2 trimer is initially formed, and, subsequently, trimers associate to form the assembly[30]. In addition, van der Waals (vdW) interactions are known to correlate qualitatively with the experimental affinities of known rigid protein–protein interactions[31]. Calculated values for both interfaces corroborate the more permanent nature of the intra-trimeric interface that has three times lower vdW energy compared to its inter-trimeric counterpart (Fig. 2c).

The final 60-meric model of the core shows excellent agreement with the cryo-EM map, with a density-map cross-correlation value of 0.84 when compared to our previous model[21] with only a cross-correlation value of 0.53. In the intra-trimeric interface, a large contributor to the buried surface area is a well-resolved cavity. Inside this cavity, three arginines with a modeled distance of 3.26 Å are visible (Fig. 2d). We cannot rule

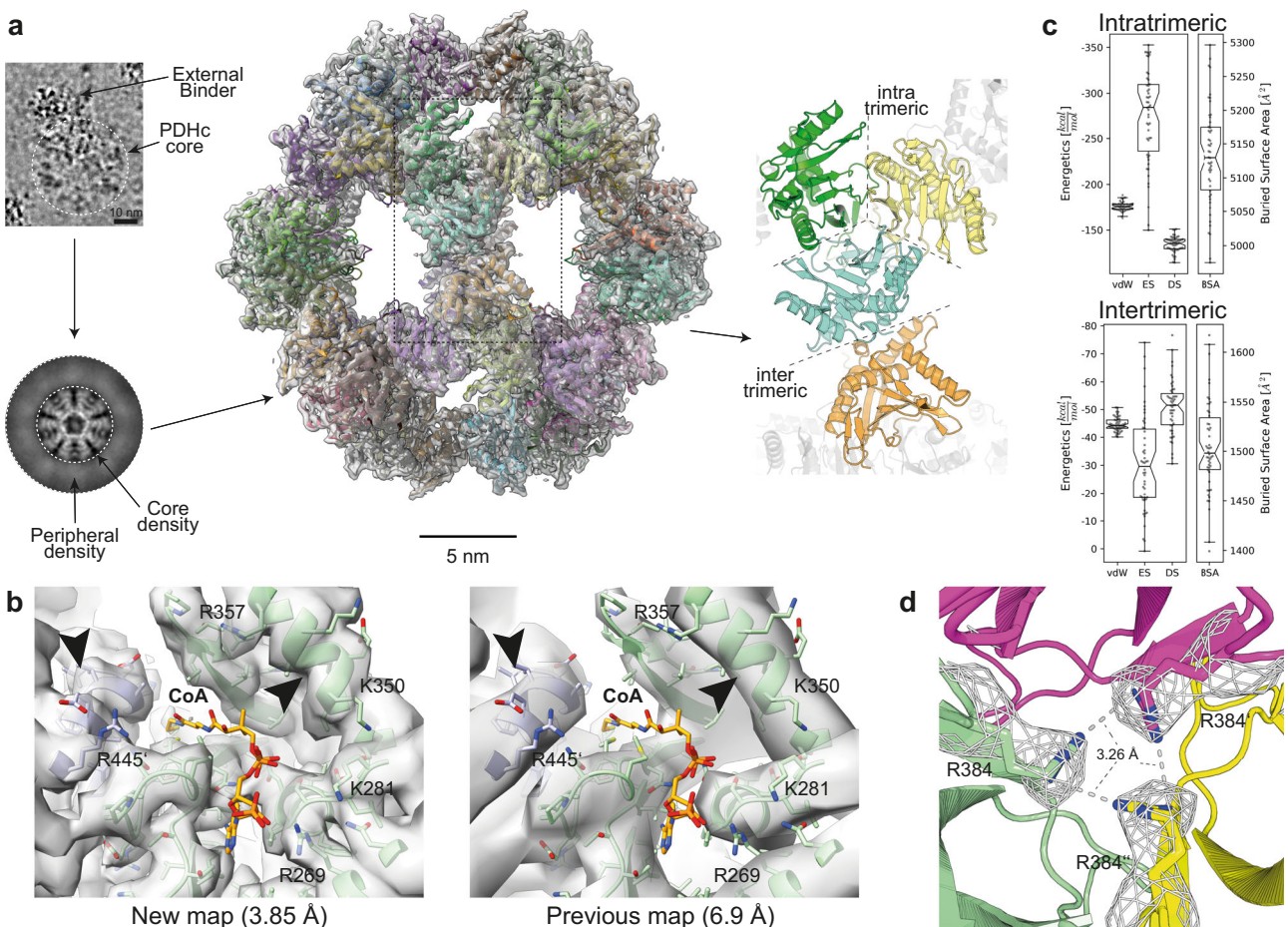

**Fig. 2 Near atomic resolution reconstruction and characterization of the PDHc core. a** Icosahedrally-averaged cryo-EM reconstruction of PDHc core. Densities corresponding to PDHc higher-order binders, visible in raw micrographs (scale bar represents 10 nm), as well as external PDHc densities of the metabolon, visible in 2D classes, are averaged to retrieve the high-resolution core structure. Each E2 monomer is colored uniquely. **b** Comparison of improvement in resolution, compared to previous data[21]. CoA molecule is not resolved but is here computationally placed and refined within the binding site as described in the Methods section. Sidechains in the CoA binding pocket of the active site are distinguishable. Helical pitch (arrowhead) allows now the unambiguous placement of the backbone. **c** Energetic calculations within the E2 core structure from $n = 50$ HADDOCK models. The intra-trimeric building block with an extensive buried surface area (BSA) is stabilized by electrostatics (ES) and van der Waals (vdW) energies, whereas desolvation energies (DS) have a minor contribution. In the dimeric inter-trimeric interaction, DS energy plays a major role, whereas vdW and ES energies have decreasing contributions. The box minima represent the 25th percentile, the box maxima the 75th percentile, the Notch indicated the data's median, whiskers extend to the minimum and maximum value inside of a 1.5 interquartile range. All data points are overlayed as a beeswarm plot. **d** Potential Arginine cluster in the intra-trimeric interface. Arg384 of each subunit are in close contact, thereby contributing to the electrostatic binding energy. Contour levels applied for the cryo-EM maps are as follows: **a** 0.05, **b** 0.041, **c** 0.03 (new map) and 0.09 (old map).

out, that this close proximity is a result of averaging, but we suspect the existence of an arginine cluster. These are very rare but may form a delocalized π-electron network, which greatly improves protein stability[32]. This result is consistent with the energy calculations previously performed, deriving the very favorable electrostatic energy contributions (Fig. 2c). This Arg is unique for Fungi, as shown by a large-scale comparative sequence analysis that we performed (Supplementary Fig. 4). In Metazoans, at this position, an Asn or His is present, while in Plants a Gln is frequently found (Supplementary Fig. 4). As these residues are also capable of forming delocalized π-electron networks[32,33], this might be a conserved feature of the PDHc core, providing it with its characteristically unusual stability.

Another structural feature is a loop element facing inside the core cavity formed by residues Val405-Trp420, and maybe potentially critical for the organization of the PDHc core scaffold. This loop exhibits high conformational flexibility, as evident by the reduced resolution at this region (Supplementary Fig. 5). Interestingly, the termini of the loop are stabilized by an extensive

hydrogen-bonding network (Supplementary Fig. 6). This indicates that, even though the loop itself exhibits a large conformational variation, it is confined to a specific space, defined by the reduced conformational flexibility of its anchor points.

**Structural analysis of the E3BP and identification of an evolutionary conserved minimal fold structuring the PDHc core scaffold.** Up to now, the structure of E3BP in the context of the PDHc metabolon is unknown. Previously, E3BP has been localized inside the yeast PDHc core[34], observed at lower resolution in the *N. crassa* core scaffold[20], and recently, identified in the *C. thermophilum* native core scaffold[21]. All these previous works hypothesized the E3BP localization and derived its stoichiometry but proposed limited models for how it might structurally organize.

Here, we have reconstructed the PDHc core scaffold at a resolution sufficient to extend the molecular model of

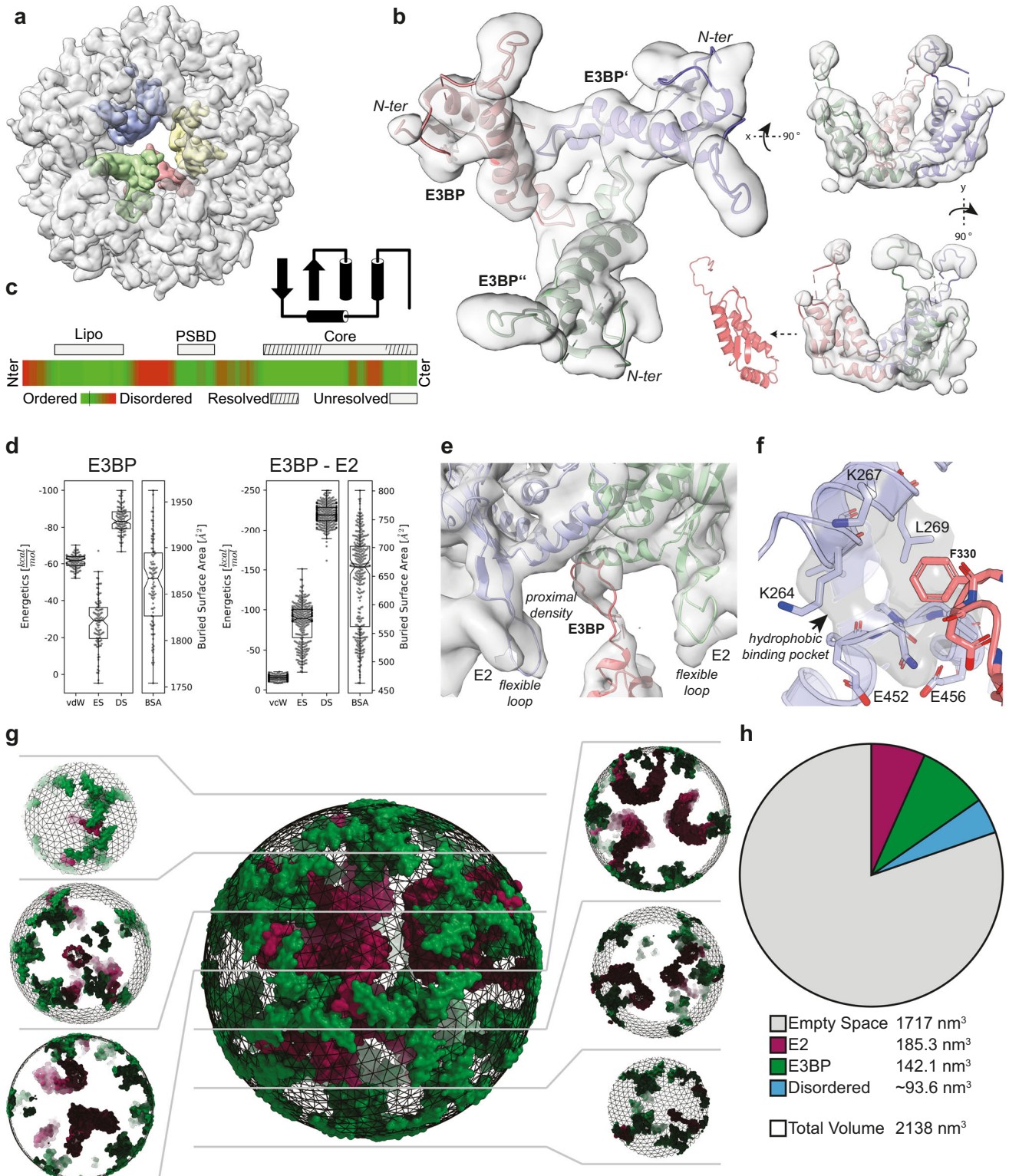

*C. thermophilum* E3BP (6.21 Å, C2 symmetry, FSC = 0.143) (Fig. 3a, b and Supplementary Fig. 7). E3BP is bound inside the icosahedral core with tetrahedral symmetry, with a center-of-mass distance of ~70 Å. Each of these four densities consists of three E3BP monomers (Supplementary Fig. 8b). Overall, the sequence organization of E3BP is highly similar to E2, having an *N-ter* LD, a central peripheral subunit binding domain (PSBD), and the *C-ter* core region connected by disordered linkers (Fig. 3c).

Results show that the core structural elements of E3BP form a minimal fold (Fig. 3c), which is conserved and can be found in various other E2 proteins from diverse acyl-transferases, including the 2-keto acid dehydrogenase family (Supplementary Fig. 9). The minimal fold is, consequently, implicated in different assembly modes that include trimers (e.g., E3BP), octahedra (e.g., E2 from OGDHc, E2 from BCKDHc, E2 from bacterial PDHc), and icosahedra (e.g., E2 from eukaryotic PDHc) (Supplementary Fig. 9). This minimal fold is composed of a β-β

**Fig. 3 Localization of E3BP assembling the PDHc core scaffold. a** Localization of four distinct densities inside the core structure, visible in the C2 reconstructed cryo-EM density map. **b** Atomic model of the E3BP trimer in one of the four recapitulated densities shows a prominent fit. **c** Domain organization of *C. thermophilum* E3BP. The primary sequence is strikingly similar to E2, consisting of an LD, a PSBD, and the core region, separated by flexible linkers predicted as disordered, analyzed by SPOT-Disorder2[68]. The resolved core structure of E3BP is composed of a β-β motif covering the N- and C-terminal regions of the fold. The unresolved central region is not seen in the reconstruction, corroborating its disordered nature. **d** Energetic calculations of the trimeric E3BP from $n = 50$ HADDOCK models. E3BP is mainly stabilized by DS, and the E2–E3BP interaction, composed of two E2s and one E3BP is primarily stabilized by DS. The box minima represent the 25th percentile, the box maxima the 75th percentile, the Notch indicated the data's median, whiskers extend to the minimum and maximum value inside of a 1.5 interquartile range. All data points are overlayed as a beeswarm plot. **e** Interaction interface of E3BP and E2. A proximal density near the inter-trimeric interface of the core, identified as an extension of E3BP is located between two flexible loops of two E2s from two different trimeric building blocks. **f** Interaction interface formed by E3BP and E2. Phe330 of E3BP is buried in a hydrophobic pocket formed by E2 apolar residues. **g** Volumetric analysis of the PDHc core. Displayed is a sphere with a radius of 80 Å representing the inner part of the PDHc core scaffold. This includes the E3BP (purple) and the internal region of E2 (green) but excluding the majority of the core outer structure. Inlets left and right show slices of the core with a sequential spacing of ~13.5 Å, highlighting the occupancy of polypeptide chains within the core scaffold. **h** Pie chart representing inner occupancies for the PDHc core scaffold. Although most of the core is calculated as empty, a substantial ~20% of the total volume is actually filled with polypeptide chains. Contour levels applied for the cryo-EM maps are as follows: **a** 0.035, **b** 0.035, **e** 0.03.

motif that is connected via a specific loop and α-helical elements (Fig. 3b, c). Based on the identified homologs, we constructed the most complete E3BP *C-ter* model to date, which covers distinct N- and C-terminal regions within the E3BP sequence (Fig. 3b, c). As expected, the core region predicted as unstructured (residues Pro364–Glu405) was completely absent from both lower-symmetry reconstructions (C1 and C2) indicating, indeed, a flexible region.

Energetic calculations of the real space refined trimeric assembly of E3BP showed that the complex has a relatively large buried surface area (~1900 Å$^2$) and is mainly stabilized by desolvation and van der Waals energies, while electrostatic energies play only a minor role (Fig. 3d). This is contrary to the E2 core assembly, which is primarily driven by electrostatic energies and probable local delocalized π-electron networks. This extensive buried surface area represents a trimeric interface of notable stability.

**The PDHc core scaffold is mediated by a weak interaction interface between E2 and E3BP and exhibits localized dynamics.** An additional density was observed that belongs to the E3BP, interacting directly with the E2 core (Fig. 3e). This density was used to discover the interaction interface between E3BP and the E2 core. Overall, E3BP is bound in a tripod-like manner and the interaction to the E2 core is primarily mediated by a loop region of E3BP (residues Pro317–Leu336). The E3BP loop region is partially buried at the inter-trimeric interface of the E2 core and flanked by three E2 loop regions which exhibited high flexibility (Supplementary Fig. 5 and Fig. 3e). Energetic calculations revealed that this interaction is substantially driven by desolvation energies (Fig. 3d), which are connected to the hydrophobic effect[35]. Mainly responsible, therefore, is the binding of Phe330 from E3BP in a hydrophobic pocket in the E2 core (Fig. 3f).

To further understand the inner core architecture, we calculated the volumetric capacity of E3BP, including both the resolved structure and volumetric estimates for the flexible regions. Surprisingly, the inner volume of the E2 core is filled up to 25% (Fig. 3g, h), mostly by the ordered E3BP minimal fold and the inner loops of the E2. The flexible region within the E3BP core region occupies ~5% of the volume and is locally confined in near-symmetric tetrahedral occupancies, in proximity to the symmetric E3BP minimal fold, close to the E2.

**Implications of the native PDHc core scaffold for the higher-order assembly and function of the complete metabolon.** E3BP is responsible for tethering E3 in proximity to the core, forming the entire metabolon. The number of E3s is limited by the number of E3BPs, and our observed model of 12 E3BPs inside the core is in agreement with previous observations[20,21]. For a full PDHc reaction to occur, the E3 has to be in relative proximity to the other PDHc component enzymes. Therefore, we postulate that E3BP cannot only have a structural role, but its localization must depend on the capacity of the E2 active site to accommodate the acetylated LD. To scrutinize this hypothesis, we reconstructed the PDHc core scaffold asymmetrically to a resolution of 7.62 Å (FSC = 0.143, Supplementary Fig. 10). The map allowed clear placement of the PDHc core scaffold within the resolved asymmetric densities. After placement, we calculated all distances between the 60 E2 and 12 E3BP monomers (E2$_{Asn251}$–E3BP$_{Ala271}$). Results reveal preferential distributions underlying the E2–E3BP vicinity (Fig. 4a). In detail, although most partners are within distances of 55–70 Å, others are further apart (>75 Å) (Fig. 4b). Distal E2s are located in a tetrahedral manner above the triangular plane formed by the E3BPs (Fig. 4a).

Since the map was asymmetrically reconstructed, additional, distinct densities above the E2 active sites appear at lower thresholds (Fig. 4c), correlating with the presence of the backfolded element which is not part of the active site[36]. This element is, however, resolved in other E2 core structures[20,22,37] and obstructs the transient binding of the LD. To estimate if this element is recapitulated in the asymmetrically-resolved native PDHc core scaffold reconstruction, we systematically altered the map density threshold and calculated the number of atoms that are excluded from the emerging low-resolution density (Fig. 4d). We observe an asymmetric coverage of the map density from distinct E2 monomers (Fig. 4d, e), which translates into localized presence or absence of this backfolded element (Fig. 4e). E2s that are proximal to E3BPs (Group 1) are less likely to exhibit additional densities above their active sites (Fig. 4e). Distal E2s from E3BPs are observed to have increased likelihoods for the presence of additional densities (Group 2, Fig. 4d). These results show that the proximity of E3BP to the E2 is correlated with the absence of additional E2 densities, an observation that assigns a previously unknown role of E3BP.

**Insights into the transacetylase reaction catalyzed by the E2 core.** With a complete, native PDHc core scaffold, the active site of E2 can be further investigated, since it is captured in a state where the backfolded structural element is, in most E2 monomers, absent. The LD is acetylated at the active site of E1, which is more than 100 Å away from the active sites residing in the core (Fig. 5a and Supplementary Fig. 1)[21]. The E2 active site binds CoA and catalyzes the transacetylase reaction, in which the acetyl group from the N6-lipoyllysine, located in the peripheral LD of

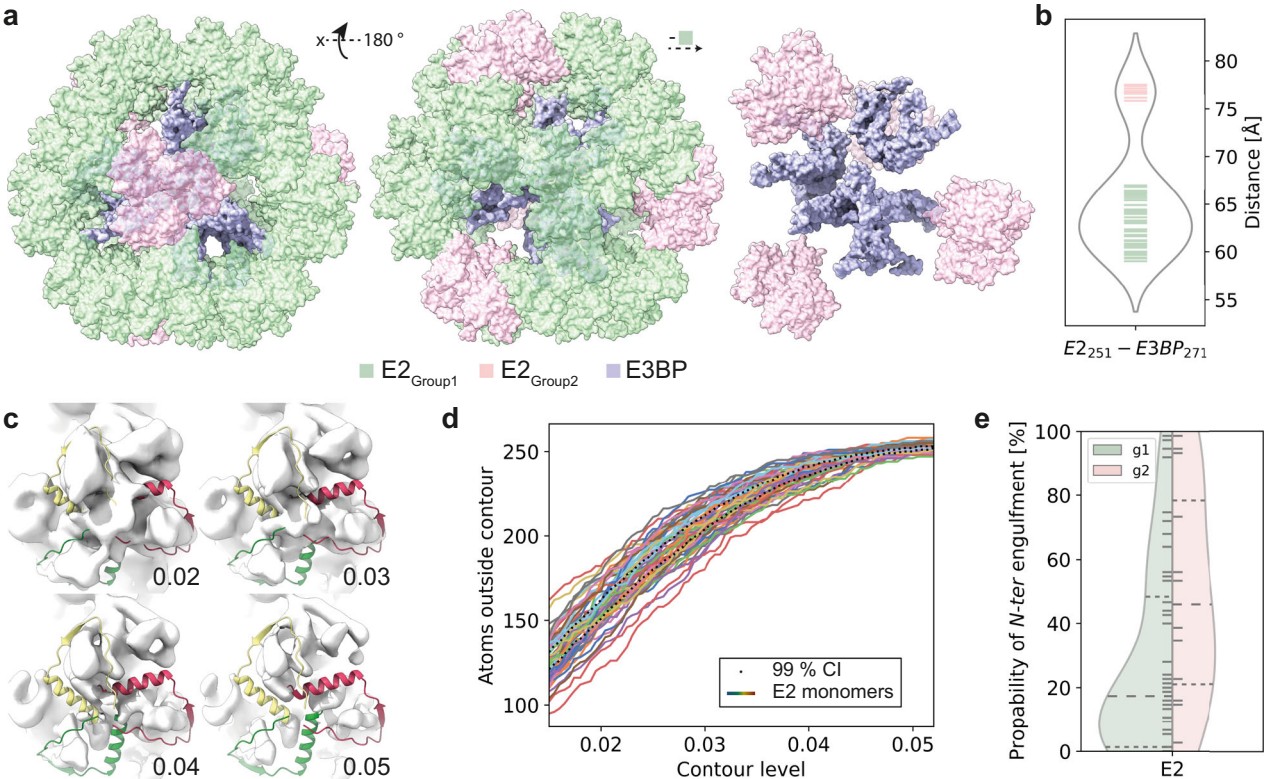

**Fig. 4 Structural impact of E3BP localization on the E2 core. a** Binary classification of E2 proximity to the E3BP. Proximal E2 monomers are colored green, whereas distal are colored pink. **b** Violin plot showing the distribution of distances of each E2 to the closest E3BP. Two groups are retrieved: Group 1, in the range of 55 to 70 Å, and Group2, in the range of more than 75 Å. **c** Enclosure of the backfolded element corresponding to the *N-ter* region of the E2 at different map contour levels (0.02–0.05). **d** Number of atoms from the *N-ter* backfolded element not covered by the cryo-EM map density as a function of applied contour level. The 99% confidence interval (CI) is displayed as dotted lines. **e** Grouped violin plot showing the probability of additional density to encapsulate the *N-ter* backfolded element in the two distinct Groups. Group 1 (g1), represents vicinal E2 monomers to E3BPs that have a substantially lower probability to exhibit densities that may cover the *N-ter* backfolded element. Dashed lines indicate lower and higher quartile and median values.

E2 and E3BP, is transferred onto the thiol group of CoA (Fig. 5a). The LD is afterward reoxidized at the active site of E3, which is again more than 100 Å away from the E2 core active site[21].

To analyze the central transacetylase reaction in the native PDHc core scaffold, we performed docking calculations, where the unbound LD and the PDHc core scaffold were considered as interacting partners. As a distance restraint, the length of the *N6*-lipoyllysine was used during flexible docking with HAD-DOCK. Although various clusters appear, the major, top-ranking cluster showed optimal energetics and, overall, more favorable scoring as compared to others (Fig. 5b, c and Supplementary Fig 11a). In this cluster, the top-ranking solution does not only satisfy the applied distance restraint but also reveals unprecedented information regarding the formed interface (Fig. 5b). Also, this is the sole cluster that had the Lys75 correctly positioned over the lipoate binding pocket (Supplementary Fig. 11b). By superimposing the *N6*-lipoyl modification on the Lys, forming the lipoyllysine (LA2), no obvious clashes in the active site were observed (Supplementary Fig. 11c). The *N-ter* of the three monomers, located above the intra-trimeric interface of the trimeric E2 building block, are freely accessible. This indicates that there is no simultaneous occupancy of the docked LD and the unresolved flexible linkers, independently corroborating the reliability of the docking calculations (Fig. 5b, c). Not explicitly considering unstructured regions during docking calculations may limit the interpretation of the overall complex stability. In particular, additional disordered regions interacting with non-interface surface parts may well regulate the stability of the complex. Summarizing, the selected cluster satisfies all known properties of

a bound LD, representing a likely solution of the transient interaction.

In detail, the LD is directly interacting with a single E2, and the binding interface is formed by three E2 loops, spanning Pro300–Asp315, Lys360–Thr370, and Phe385–Ala395 (Fig. 5d). The interaction between the core and the docked LD is stabilized by strong electrostatic energies (Fig. 5c), clearly supported by the ionic interaction within the E2–LD interface, as formed by E2 residue Arg307 and LD residue Asp79 and Glu81 (Fig. 5d). Strong electrostatic complementarity also drives various other carrier proteins in multienzyme complexes where swinging arms are involved in substrate channeling[24]. The docking calculations were then repeated for the published human and *N. crassa* E2 models (PDB ID: 6CT0[37] and PDB ID: 6ZLM[20]) and their corresponding LDs. In both E2 core structures, a region corresponding to the *N-ter* of E2, that we observed in the native PDHc metabolon to be unstructured[21] or of low occupancy (Fig. 4) is backfolded onto the core, forming an extensive fold with a large surface area thus reducing the active site accessibility. The docking results for both simulations shows that the lipoylated lysine cannot access either the human or the *N. crassa* E2 active site due to shielding stemming from the polypeptide chain of the backfolded element (Supplementary Fig. 12). These structures, therefore, must represent states of the E2 core that are not expected to accommodate the LD in the E2 active site.

We selected the top-ranking docking model for subsequent analysis via molecular dynamics (MD) simulations to evaluate the stability of the modeled E2–LD complex (Supplementary Fig. 13). The system included both the lipoate and the CoA.

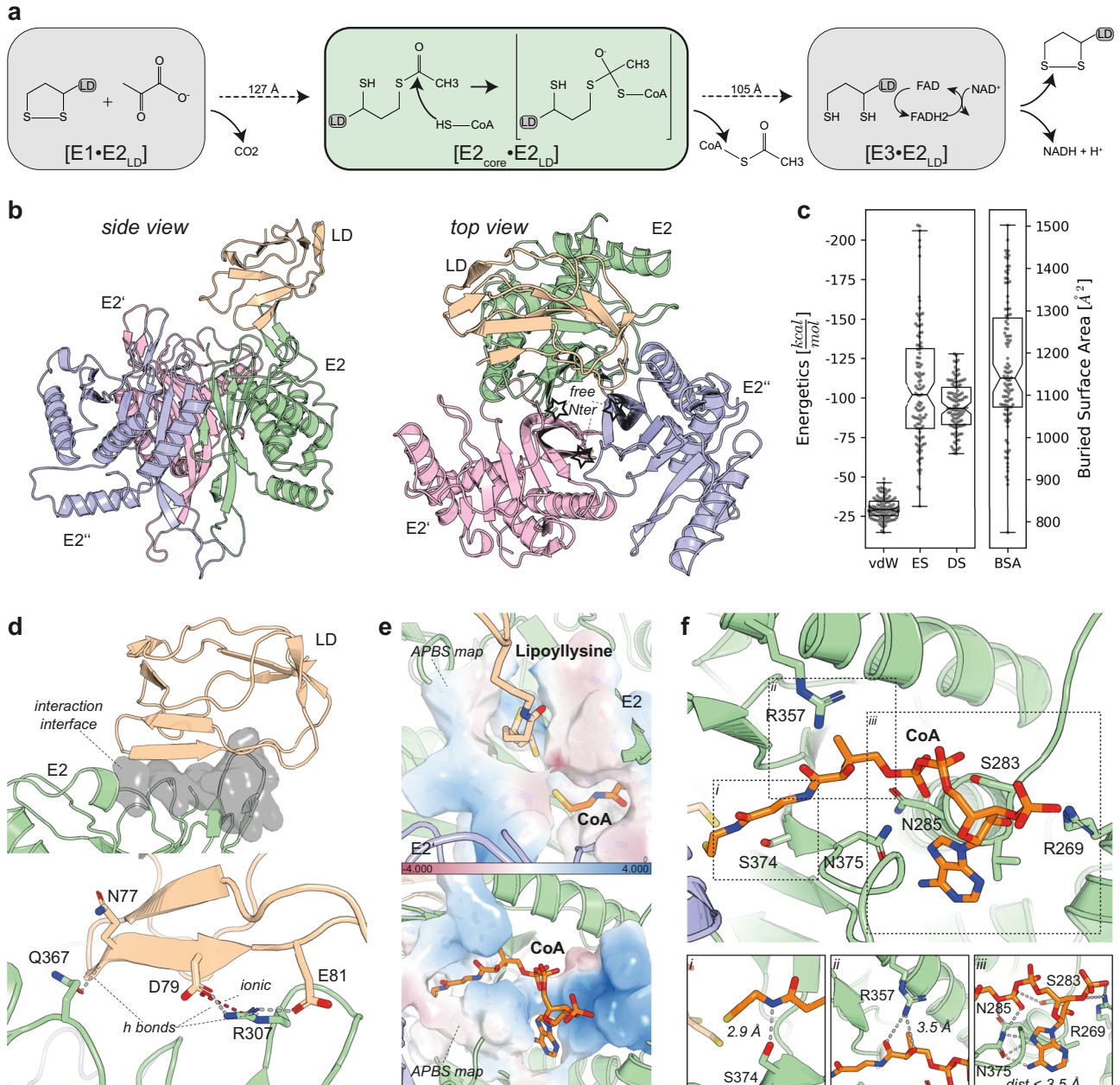

**Fig. 5 Structure-based analysis of the transacetylase reaction within the PDHc core scaffold. a** Schematic representation of the reaction steps implicating the lipoate moiety of the LD in the complete pyruvate oxidation reaction. The focus is on the highly transient E2–LD interaction. **b** Top-scoring solution from the top-scoring cluster derived from the docking simulations (Supplementary Fig. 9) that included the LD and the E2 core. The *N-ter* (☆) of all three E2s is freely accessible in the derived docking model. **c** HADDOCK energetics of the selected cluster, containing $n = 104$ models. Binding is mainly mediated by ES and DS, while vdW contribution is decreased. The box minima represent the 25th percentile, the box maxima the 75th percentile, the Notch indicated the data's median, whiskers extend to the minimum and maximum value inside of a 1.5 interquartile range. All data points are overlayed as a beeswarm plot. **d** Interaction interface between LD and E2. The interaction is stabilized by three H bonds and a very strong ionic interaction between the Arg307 of E2 and the Asp79 of LD. **e** Electrostatic surface potential maps of the two substrate-binding pockets in the active site. The lipoate binding pocket is overall uncharged, whereas the CoA binding pocket stabilizes the negatively charged phosphates. **f** Binding of the CoA in the active site. Inlets i–iii show the hydrogen-bonding networks stabilizing the complete molecule. The lipoate and CoA were derived from docking procedures and refined during MD simulation (Supplementary Fig. 13, see Methods).

After the relaxation of the system, we were able to further visualize the active site of the transacetylase when the LD is bound. Calculations of electrostatic surface potential maps of the binding pockets for the acetylated lipoate in particular showed an overall uncharged binding cavity (Fig. 5e). This is in contrast to the interface rim which is controlled by high electrostatic energy (Fig. 5c). On the other hand, in the CoA binding pocket, extensive

areas of diffuse positive charge are visible, accurately defining an interaction surface for the negatively charged phosphates of CoA to be nicely accommodated (Fig. 5e). In detail, the CoA in *C. thermophilum* E2 active site is interacting with six distinct residue side chains (Fig. 5f). All involved residues are highly conserved (>96% conservation rate) among all eukaryotes, except Arg357 (76%), which is substituted by Lys (23%), a commonly

considered functional mutation. The cryo-EM data for the symmetric reconstruction of the PDHc core scaffold show decreased resolution for secondary structure elements surrounding the CoA binding site (Supplementary Fig. 5), possibly participating in a CoA-gating mechanism, similarly described in other CoA-binding enzymes[38]. Overall, we derived a previously unknown architectural model describing the E2–LD interaction, therefore, structurally mapping the native, metabolon-embedded transacetylase reaction.

## Discussion

Visualization and characterization of large protein assemblies directly from the native source is becoming increasingly possible due to the recent advances in cryo-EM[39]. Here, we have optimized our previously established single-step fractionation protocol[21] to enrich for the endogenous, active 10 MDa PDHc metabolon, a critical complex involved in primary metabolism that performs "the link reaction". Even though the studied fraction is of notable complexity, including many other complexes, structural characterization of PDHc is feasible due to advances in image processing algorithms that allow discriminating chemically heterogeneous specimens of varying degrees of flexibility[40]. Also, the chosen pixel size of 1.567 Å is preferable for this kind of sample. The magnification used is the major resolution limiting factor considering that ~85% of the Nyquist frequency is the maximum that is realistically achievable with our cryo-EM equipment. However, with higher magnification, the average PDHc particle per micrograph would be reduced from ~3 to ~0.7, which disproportionately increases microscope acquisition time and downstream analysis. The recent advances in cryo-EM, especially the employment of faster direct electron detectors could tackle this issue by acquiring more data of higher quality in the same time frame, leading to an overall higher resolution of metabolon structures in low abundance—even when highly heterogeneous samples are studied.

Unambiguous placement of residues at 3.85 Å resolution is possible for the majority of backbone atoms but limited for the side chains. Bumps at the backbone trace indicate side chains, therefore Cα and Cβ atoms can be placed, but exact side chain conformations are not often resolved. To address this issue, we applied integrative structural methods. During real-space refinement and HADDOCK energetic calculations, simulated annealing cycles are performed informed by rotamer libraries to improve underlying stereochemistry (e.g., side chain interactions, preferred rotamer conformations, etc.).

One of the completely resolved side chains is the Arg384 in the inside cavity. Due to this high local resolution and the short distances between the modeled guanidine groups, we hypothesize the presence of a delocalized π-electron network, which would increase the core scaffold's stability. We cannot rule out, that this derived proximity could be a result of averaging, indicating that the arginine side chains can be present in different rotamers forming other interaction networks across interfaces.

To our knowledge, reaching details reported in this manuscript is unparalleled for localized compartments of any studied endogenous metabolon to date, and in particular of PDHc. By annotating the complete PDHc core scaffold, which is structured by folded domains of E2 and E3BP, we observed that various protein regions are not clearly visible, pointing to an intricate role of localized flexibility. While it is known that the LD, the E1, and the E3 are organized via flexible tethers, we provide a rationalization for further flexible regions present in the PDHc metabolon. Flexible regions were first detected above the E2 active site. A previously described backfolded structural element is not seen in the native PDHc core scaffold structure, allowing increased accessibility to the LD to be accommodated in the E2 active site. We observed that this element that occludes the accessibility of the active site is only captured at lower density thresholds, only when the PDHc core scaffold is asymmetrically reconstructed and only in E2 subunits that are not in proximity to E3BP. This observation reveals an interplay of localized flexibility: Above the E2 active site, the E2 *N-ter* must be in a flexible state to accommodate the acetylated LD, while inside the PDHc core, where E3BP is present, additional flexible regions of E3BP are in proximity of the CoA binding site. These flexible regions inside the PDHc core scaffold stemming from the E3BP are not only frequently proximal to the CoA binding site, but also to E2 active sites that are able to accommodate acetylated LD. Although CoA availability in mitochondria is very high[41,42], and CoA can in principle directly diffuse in the CoA-binding site, an additional level of diffusion regulation via localized unstructured regions from E3BP could well be possible. We cannot exclude that these *N-ter* disordered regions, of both E2 and E3BP may play a role in stabilizing and properly orienting LD for catalysis after the LD is bound. In agreement with our cryo-EM reconstruction, the *N-ter* of the E2s are in a flexible, unstructured state and, therefore, not considered during the docking calculations. However, enough empty volume is observed in the docking solutions at the interface periphery to accommodate flexible regions that possibly regulate the transacetylase reaction.

Our data suggest a binding mode for the E2 core scaffold–LD complex where LA2 is inserted in the active site, unhindered, satisfying applied restraints in a conformation where the reaction distance to the CoA is indeed plausible. This suggests that the proposed mechanistic model either represents an encounter state or a relatively stable bound conformation. In the native, endogenous, and active eukaryotic PDHc, element Q227-N251 is unfolded, as shown by the presented cryo-EM data. If these flexible regions are modeled in a helical conformation, then minor, side chain clashes may appear in the proposed model. Although this helix is not part of the E2 catalytic domain[36], it was recently suggested to aid the stabilization of the E2–LD complex in the octahedral *E. coli* E2 counterpart in the absence of pyruvate[43]. Considering the high local sequence similarity, it is possible that disorder-to-helix transitions may additionally occur, shifting the LD by ~6 Å relative to the E2 core and, presumably, stabilizing the encounter complex at any timepoint during CoA acetylation.

Another interesting aspect of the native, active PDHc core scaffold is the interplay of non-covalent forces among discovered protein–protein interfaces and active sites. It is observed that structural interfaces within PDHc are rather large, and regulated by van der Waals and desolvation energies. However, transient interfaces and the E2 active site employ other combinations of non-covalent interactions, including significantly smaller interface areas and complementary electrostatics. The LD is attracted by an exposed positively charged surface of the E2 and extends the lipoate within a dedicated hydrophobic cleft. A similar electrostatic complementarity mechanism has been observed for other large enzymatic assemblies, especially those with swinging arms[24], and in particular in biotinylated[44] and acetylated carrier domains[45] and also witnessed for acetylated carrier domains encountered in fatty acid synthesis within native cell extracts[46]. It is indeed interesting to postulate that swinging arms in multi-protein complexes have independently evolved to take advantage of their highly dynamic nature: Swinging-arm-mediated substrate shuttling is underlined by a steering force of complementary electrostatics that is triggered when the respective enzymatic sites are close.

Future research on the endogenous PDHc metabolon can more broadly shed light on how these systems may potentially evolve in

different species, which is still a topic not completely understood. Furthermore, deeper insight into PDHc metabolons and especially a more accurate description of the locations of E1 and E3 proteins in the outer shell of the native PDHc metabolon can help make strides toward advancing the knowledge in modeling the complete pyruvate oxidation reaction. To further characterize the structural intermediates of the PDHc catalyzed reaction, excess of substrates or the design of substrate analogs, which could trap the metabolon at specific points, could be used to shift the equilibrium. Additionally, future hypotheses based on the docking and interaction energetic calculations presented in our study can open the doors to site-specific mutagenesis experiments with the aim to regulate underlying binding affinities. Such experiments, however, cannot be performed in cellular homogenates; instead, reconstitution of pure components of the giant PDHc metabolon must be performed to ultimately control and fine-tune the reaction conditions.

By presenting a high-resolution structure of the endogenous PDHc core scaffold, our work provides key details toward achieving a greater understanding, not only of the gigantic PDHc metabolon but also of α-keto acid dehydrogenase complexes and other highly complex enzymatic systems regulated by swinging arm mechanisms of substrate transfer.

## Methods

**Cell cultivation and lysate fractionation.** The general procedure[21] was applied with minor adaptations, described below in detail. For optimal PDHc yield, cells should preferably be in the log phase. To achieve this, the liquid culture was passaged carefully every 2 days of growth, eventually leading to a sufficient number of cells. The log phase is indicated by the equal size of mycelia (~1 cm in diameter) and clear surrounding media. At this stage, oxidative metabolism is fully active and high yields of PDHc could be retrieved. At later stages, mycelia with different sizes occupy the flask and new small cells are filling the media, leading to a turbid appearance. During this, cell cycle arrest and most likely anaerobic metabolism are dominant, which could lead to very low PDHc yields. For cell lysis, we increased the bead-beating cycles from three to six, which further increases the protein concentration in the cleared lysate, indicating previously incomplete lysis of *C. thermophilum* cells.

**Pyruvate dehydrogenase (PDHc) activity assays.** The protocol to assay for PDHc activity was adapted from ref. [47]. The reaction mixture (100 μL) contained 30 mM potassium phosphate (pH 7.5), 100 mM NaCl, 2 mM MgCl₂, 3 mM NAD⁺, 5 to 1000 μM CoA, 2 mM ThDP, 4 μL Cell Counting Kit 8, and 2 μL cell lysate containing PDHc. The reaction was preincubated for 5 min at 37 °C and initiated by titrating concentrations of pyruvate (0.1 to 20 mM) and spectrophotometrically following the formation of the Formazan product of Cell Counting Kit 8 at 460 nm. Product formation was calculated using the Lambert–Beer-Equation and the ε value for the formazan product of $3.07 \times 10^4$ L mol⁻¹ cm⁻¹ [48].

**Immunoblotting experiments.** In-house casted gels were prepared for the Western Blot experiments using: 10% (w/v) acrylamide (37.5:1), 370 mM Tris-HCl pH 8.8, 0.1% (w/v) sodium dodecyl sulfate (SDS), 0.04% (w/v) APS, 0.002% (v/v) TEMED in ddH2O for the separating phase and 5% (w/v) acrylamide (37.5:1), 125 mM Tris-HCl pH 6.8, 0.1% (w/v) sodium dodecyl sulfate (SDS), 0.04% (w/v) APS, 0.002% (v/v) TEMED in ddH2O for the stacking phase. The thickness of all gels was 1 mm and they were freshly prepared before use. Samples were incubated for 5 min at 100 °C mixed with a 4x loading dye (250 mM Tris-HCl (pH 6.8), 8% w/v SDS, 0.2% w/v bromophenol blue, 40% v/v glycerol, 20% v/v β-mercaptoethanol). About 400 ng protein per fraction and 8 ng recombinant protein was loaded in each lane accompanied with 5 μl of Precision Plus Protein™ All Blue Prestained Protein Standards (Biorad #1610373) for each set of samples. Polyacrylamide gels were electrophorized in a 1X electrophoresis buffer freshly prepared from a 10X stock (30.3 g Tris-base, 144 g Glycine in 1 L of deionized water), at an electrical field of 100 V for ~2 h. After this step gel contents were transferred (in pairs) to a nitrocellulose membrane using a Trans-Blot® Turbo™ Transfer System of BioRad in a pre-set protocol of 25 V (1 A) applied field, for 30 min. Following the transferring step membranes were blocked in 5% w/v milk/TBST solution for 1 h under stirring before being incubated for 16 h at 4 °C with the primary antibody (Ab) that was diluted and freshly prepared in 2% w/v milk/TBST solution to a final concentration of 0.2 μg/ml. The next day primary Ab was removed and three washing steps with 2% w/v milk/TBST solution were performed, before the secondary Ab (0.1 μg/ml, also in 2% w/v milk/TBST solution) was applied and incubated for 1 h. Finally, three washing steps were carried out and membranes were screened using a ChemiDoc MP Imaging system and a freshly prepared ECL fluorescent mixture under optimal exposure time conditions. Antibodies were custom made by GenScript (New Jersey, USA), using the sequences provided in Supplementary Table 1.

**Cryo-EM sample preparation and data collection.** For cryo-EM analysis of the samples, carbon-coated holey support film type R2/1 on 200 mesh copper grids from Quantifoil® were used. Grids were glow discharged using a PELCO easiGlow™, 15 mA, grid negative, at 0.4 mbar and 25 s glowing time. A 3.5 μL volume with a total protein concentration of 0.3 mg mL⁻¹ was applied on each grid which was then plunge-frozen using the Vitrobot® Mark IV System (Thermo Fisher Scientific), after blotting with standard Vitrobot Filter Paper (Grade 595 ash-free filter paper ⌀55/20 mm). Conditions in the chamber during the whole process were kept stable at 4 °C temperature and 95% humidity. For plunging, blot force of 2 and blotting time of 6 s were applied. The vitrified grids were clipped and mounted onto a Thermo Fisher Scientific Glacios 200 kV Cryo-transmission electron microscope under cryo and low humidity conditions. Images were acquired using Falcon 3EC direct electron detector in linear mode and a total dose of 30 e-/Å². The beam was aligned to be parallel and perpendicular to the sample having 2.5 μm diameter and the objective angle was restricted by a 100 μm objective aperture. The acquisition parameters and further cryo-EM and structure statistics are shown in Supplementary Table 2.

**Image processing.** About 2810 movies acquired at low dose were imported in RELION 3.1[40]. Movies were motion-corrected with the in-software implementation[49], and CTF calculation was performed with gctf[50]. Particle picking was manually performed to identify PDHc, and then template-based picking was applied, yielding a total of 296779 particles. After sequential 2D and 3D classifications to remove low-resolution class averages retrieved, a final dataset of 10249 single-particles was selected for structure calculations. Refinement of the maps was performed by applying icosahedral, C2, and C1 symmetries. FSC calculations were performed within RELION 3.1[40] and reporting of resolutions is according to the gold-standard FSC[51]. Local resolution estimation was performed in cryoSPARC 3.2[52].

**Model refinement.** For refinement of the E2 structure, the initial model (PDB ID 7BGJ)[21] was fitted into the cryo-EM density using ChimeraX[53] and then refined using iterative manual refinement with Coot[54] and real-space refinement with Phenix[55] with standard parameters. The previously proposed model of E3BP deposited in SBGrid was manually extended using Coot using the C2 symmetrized map and real space refined using Phenix as performed for the icosahedrally-averaged equivalent. Electrostatic potential maps, whenever reported, were calculated using the APBS plugin in PyMOL 2.4.

**HADDOCK.** For binding energy calculations of protein complexes, the HADDOCK multi-body-interface[56,57] was used. Each protein was defined as a unique chain and the docking settings were adapted for refinement purposes as described here[58], but using 50 models in all three refinement steps (it0, it1, water) instead of 20. All other parameters and subsequent clustering analysis were kept as default.

**Calculation of map cross-correlation coefficients.** For calculation of map cross-correlation coefficients the PHENIX tool "Comprehensive validation (cryoEM)" was used with default parameters[59].

**Large-scale sequence analysis.** For large-scale eukaryotic kingdom-wide sequence analysis of E2 proteins, a database with 2460 UniProt entries was built as follows: The sequences of the E2 protein of *C. thermophilum* (ID G0S4X6), *H. sapiens* (ID P10515), *S. saccharomyces* (ID P12695), *D. melanogaster* (ID Q9VM14), and *A. thaliana* (ID Q5M729) were used as a template for blastp. The first 1000 hits of each sequence were selected and merged and duplicates were then removed. All resulting 2460 sequences were aligned using Clustal Omega and sequence conservation was calculated by isolating the specific column in the alignment, based on the *C. thermophilum* sequence.

**Volumetric analysis and occupancy estimations and relevant statistical information.** To get the volume of the proteins in a sphere with a radius of 80 Å inside the core, measurement with ChimeraX[53] were performed as follows: After calculating the solvent excluded surface with a probe radius of 1.4 Å, volumes were reported using the ChimeraX function "volume". The volume of the disordered part of E3BP, which was not included in the model, was estimated using 22566 de novo generated structures. 5000 models were generated using Xplor-NIH[60], and 17566 using standard parameters in MODELLER[61].

For the volumetric analysis related to the coverage of the E2 N-ter region by the densities from the C1 reconstructed cryo-EM map of the PDHc core scaffold, calculations were performed as follows: The homology model of the *C. thermophilum* E2 with the backfolded element was fitted and optimized into the C1 density 60 times, forming the complete biological unit. The enclosure of each E2 monomer was calculated by ChimeraX using the *fitmap* command with *maxSteps* set to 0 for map contour levels corresponding to a minimum of 0.015, a maximum of 0.052, and a step-size of 0.0005. The confidence index (CI) per contour level for

all 60 monomers was calculated by $CI = \mu \pm 3.291 \frac{\sigma}{\sqrt{60}}$, where μ is the mean and σ is the standard deviation. The probability of an enclosed element per E2 was calculated using $p = \frac{n}{N}$, where $n$ is the number of data points below the confidence range and $N$ is the total number of contour levels probed ($N = 75$). Data were treated following a normal distribution.

**Macromolecular, flexible docking of LD domain**. For docking of the LD domain to the core scaffold, the homology model of the LD domain[62] was used. For the human ortholog, the published structure of the LD was used (PDB ID 1Y8N)[63], for *N. crassa*, a homology model based on the human model was generated using MODELLER by default. The pentameric building block of the E2 core was used, targeting the E2 core trimer to avoid retrieving solutions that would violate the architecture of the core assembly. For *C. thermophilum*, the here presented refined core structure was used (Supplementary Table 2). For the human and *N. crassa* equivalents, the corresponding published structures were used[20,37], respectively.

As an experimentally-derived distance restraint, we incorporated the distance between the Cα of the lipoylated Lys75 in the LD (Lys259 in humans and Lys75 in *N. crassa*) and the Cβ of Ala387 (Ala578 in humans and Ala386 in *N. crassa*). This distance was considered to span 10 to 25 Å, explicitly accounting for the lipoylated Lysine and the proximity to the CoA. Docking was then performed using HADDOCK 2.2 guru interface by default, but an increasing sampling, retrieving 10000 models in *it0*, 1000 models in *it1*, and 500 models in water refinement (wat).

**Model modifications**. Prior to the MD simulations, CoA and LA2 were added to the model. For the CoA molecule, the CAO from the PDB ID 1EAD was superimposed and the derived pdb file was modified by removal of the O1 atom and relabeling to COA. The LA2 residue was added by superimposing the LA2 from the PDB ligand library onto the Lys75 and eventually connected using PyMOL. The derived modified residue was geometrically refined using Coots "Regularize zone", which arranged the lipoate side chain clash-free in the binding pocket with allowed torsion angles.

**MD Simulations**. The MD simulations of the protein complex were performed using NVIDIA CUDA acceleration modules of NAMD 2.13[64] employing CHARMM36[65] force field with an integration time step of 2 fs applying SHAKE algorithm on all bonds involving hydrogen atoms. The long-range electrostatics were treated using the particle-mesh Ewald approach[66].

Topology and parameter files for the ligands (LA2 and CoA) were generated via the CHARMM-GUI[67]. Topology and parameters files for LA2 were subsequently modified assigning atom names and atom types of the Lys segment of LA2 up to (but not including) NZ atom according to existing topology for Lys. The remaining atom names and atom types were left unchanged. The system was solvated in a water box of TIP3 water and neutralized with 150 mM NaCl, eventually comprising 169256 atoms. An implicit bond between the S atom of CoA and the terminal S2 atom of LA2 was restrained to 3.5 Å (harmonic force constant of 5 kcal mol$^{-1}$ Å$^{-2}$).

MD simulations were performed using two approaches: (1) "hard-restrained", where only atoms of cofactors, protein residues 8 Å around the cofactors and water-ion environment were allowed to move without harmonical restrains and (2) "soft-restrained", where all atoms, except the two restrained sulfur atoms of CoA and LA2, of the system were unrestrained. Each approach was repeated three times for 100 ns per replicate. Prior to each of the MD simulations, three all-atom minimization-relaxation cycles were performed. At the beginning of each cycle, the system was minimized for 10000 steps. During the first cycle, the water-ion environment and hydrogens were allowed to relax, keeping the protein and cofactors harmonically restrained. Subsequently, the temperature was incrementally changed from 0 to 310 K, relaxing the system for 1 ps per increment of 5 K with a final relaxation for 0.1 ns at 310 K. In the second cycle, for approach (1) side chains within 8 Å around (and including) CoA and LA2, as well all hydrogens, water molecules and ions were set free while all other atoms were kept restrained, while for approach (2) only protein backbone was kept restrained. The system was incrementally heated as described for the first cycle with a final relaxation to 0.1 ns at 310 K. Finally, the unrestricted system was incrementally heated as previously described and the MD simulation was performed for 100 ns at 310 K, 1.01325 bar. All necessary files to repeat the simulations are deposited at SBGrid.

**Reporting Summary**. Further information on research design is available in the Nature Research Reporting Summary linked to this article.

## Data availability
The data that support this study are available from the corresponding author upon reasonable request. The 3D maps are available at EMDB database under the accession code EMD-13066, the molecular model of E2 at the PDB database under the accession code PDB ID: 7OTT, and the computational models as well as the parameter files for the MD simulation at the SBGrid database under the accession code 848. Source data are provided with this paper.

## Code availability
All unpublished code and scripts used in this study are available upon request.

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

## Acknowledgements

We thank Dr. Theo Karamanos (University of Leeds) for assistance with the XPLOR software and thank the Kastritis laboratory members for valuable discussions. M.S. was supported by R.G.C.C. International GmbH and BioSolutions GmbH. This work was supported by the Federal Ministry for Education and Research (BMBF, ZIK program) (Grant nos. 03Z22HN23, 03Z22HI2 and 03COV04 to P.L.K.), the European Regional Development Funds for Saxony-Anhalt (grant no. EFRE: ZS/2016/04/78115 to P.L.K.), funding by Deutsche Forschungsgemeinschaft (DFG) (project number 391498659 and RTG 2467), and the Martin-Luther University of Halle-Wittenberg.

## Author contributions

F.L.K. performed the sample preparation, biochemistry, prepared cryo-EM grids, and performed screening. F.H. collected the data. Activity assays was performed by J.M. with supervision from F.L.K. C.T. developed and performed the pipeline for all data analysis and structure calculations. P.L.K. and C.T. evaluated the results. P.L.K. and I.S. analyzed cryo-EM data. M.S. implemented the molecular dynamics with supervision from C.T. Y.S. contributed to structural modeling. P.L.K. and C.T. wrote the manuscript with input from all authors. P.L.K. conceived the project, with input from C.T. P.L.K. supervised and funded the project.

## Competing interests

The authors declare no competing interests.
