## [Peer Review File · Nature Communications]

CryoEM snapshots of a native lysate provide structural insights into a metabolon-embedded transacetylase reactionReviewers' Comments:

Reviewer #1:

Remarks to the Author:

In this manuscript Tuting et al study the enzyme complex pyruvate dehydrogenase (PDHc) from *C. thermophilum* lysate. This study is a continuation of the previous study (Kyrilis et al, Cell reports 2021), which put the molecular architecture of this complex forward. The advances of the study by Tuting et al compared to the previous study are:

- Improved resolution (E2p core from 6.9 Å to 3.85 Å, holocomplex: 14.2 Å to 6.21 Å)
- Biochemical analysis: Km values of pyruvate and CoA verified for prep
- Computational analysis of complex interface sizes and energetics
- Computational model of E3BP C-ter model fitted into core scaffold.
- Computational docking is used to derive a model for catalytic function of the enzyme complex, which is probed using molecular dynamics simulations.

Overall, the manuscript appears well written and the cryo-EM analysis is thorough within the resolution claims. On the technical side this reviewer notes that some interpretations, such as side chains and Pi-electron networks, typically base on resolutions beyond the 3.85 Å achieved in this study. Ultimately, this resolution appears to be limited by the magnification chosen for this study – which is arguably not an optimal choice for such a detailed structural dissection of the complex.

This reviewer is neither a specialist on docking/molecular dynamics nor on pyruvate dehydrogenase complexes. While the approach and the resulting swinging-arm model for enzyme function appear intriguing, this reviewer misses an effort to verify aspects of this model experimentally. For example, pyruvate, CoA or analogs, if available, could be used to shift the equilibrium of the reaction to obtain the 'CryoEM snapshots' of the reaction promised in the title.

In summary, this manuscript constitutes an advance to the metabolon field. The combination of different experimental and computational methods and the developed functional model of the PDHc are appealing. However, this reviewer would suggest down-toning some of the mechanistic interpretation, which ultimately bases on rather low-resolution cryo-EM data and computational models that await further experimental validation.

Specific comments:

- Figure 2: The visibility of side chains is not compelling – which is not surprising at a resolution of ~4 Å.
- Figure 2C/D: CoA is not resolved in the cryo-EM structure and the positioning of side chains is speculative at this resolution. Taken together, the panels may be misleading.
- Pi-electron network remains a hypothesis at this level of resolution (Figure 2E), also in L. 167.
- While the resolution claim is 3.85 Å in the abstract, in figure 2 3.8 Å is mentioned. Please clarify.
- L. 151: how did the authors measure improvement of side-chain localization? Clashes? Ramachandran outliers?
- L. 151/152: how sensitive is definition of interfaces on modeling errors that will still be present for a model based on a ~4 Å EM structure?
- Figure 5: How do the authors validate the (Haddock) model?
- L. 164: cross-correlation coefficients depend on parameters such as the low pass filter used. It may be more informative to compare the 60-mer model with decoys (or the previously published model).

Minor:

- It should probably read "... the PDHc core's unusual stability."
- Minor: the text has relatively few paragraphs, which makes it a bit difficult to follow the text. Limitation to one thought per paragraph would help.
- Minor: the figure captions include description of results that this referee would have expected in the main text.

Reviewer #2:

Remarks to the Author:

The article entitled "CryoEM snapshots of a native lysate provide structural insights into a metabolon-embedded transacetylase reaction" by Christian Tuting, Fotis Kyrilis, et al. reports the 3.85 Å resolution cryoEM map for the E2 icosahedral core of the pyruvate dehydrogenase complex (PDHc) from the thermophilic filamentous fungus *Chaetomium thermophilum*. A map of 6.21 Å resolution when considering only C2 symmetry revealed map density for part of the E3BP subunits within the core, arranged in a tetrahedron of trimers. The generated model was analyzed to reveal only ~25% of the core is filled, mostly by the E3BP minimal fold and inner loops of the E2. Docking of the Lipoyl domain (LD) onto the generated protein model identified possible binding modes that would position the lipoyl group into the active site adjacent to the coA binding site. The authors report that their preparation of endogenous PDHc metabolon in a heterogeneous, high-molecular weight cell extract fraction was key to obtaining enough particles for map construction of homogenous species. Overall, the biological significance of the work is clear in the difficulty of identifying structures of PDHc, a very complex complex with multiple subunits that links glycolysis and the TCA cycle in central metabolism. The obtained structure of the PDHc core at sub 4 Å is an improvement to previous reported maps for fungi PDHc cores, as the current maps show better definition of helices and some density for side chains. The method for sample preparation was recently published by the author's in *Cell Rep.*, leaving the extension of map resolution and the ability to do additional analysis based on those results to be the major findings and reporting of this work. The model is overall similar to the lower resolution models in *Cell Rep.* and the complex is consistent with work by Forsberg et al. *Nat. Comm.*, but the increase in resolution to 3.85Å is notable and therefore worthy of considering for publication. The analysis is nicely performed and the article is well written. A few queries for the authors are below.

Regarding data:

The FSC (Figure S1) shows a dip around 0.18 1/Å for both the masked and unmasked maps. What is the source of this dip?

Some items that could use further clarification:

- 1.) Results section 1, lines 103-129; The details that enabled obtaining a good sample of PDHc are not clear in this section, and the methods section doesn't add much more detail. "Note that the growth state of *C. thermophilum* before harvesting significantly impacted the retrieved PDHc activity and was properly adjusted to recover optimal PDHc activity" Adjusted in what way?
- 2.) Line 125; report error for Km values
- 3.) lines 179-181; It is unclear what/where loop is positioned relative to the core. Line 175 states 'inward facing loop', but this is unclear. Inward towards what?
- 4.) Line 203; two antiparallel beta-strands would make a beta-beta motif. Are there more arranged together to warrant description of a sheet? Is this a hairpin? The beta-strands are not visible in the figure.
- 5.) Line 207; the use of 'remarkably' seems unjustified based on the sentence. Rather it is consistent or expected.
- 6.) Lines 227-228; regarding "Phe330 from E3BP in a hydrophobic pocket in the E2 core". This implies hydrophobic/VDW forces would be contributing, but this is counter to the previous sentence.
- 7.) Section lines 237-267: A link is made between proximity of E2 and E3BPs being in an 'open' state, whereas E2 at the periphery of the core, and away from E3BPs, are more likely in a closed state with a

'back-folded element' positioned over the active site and preventing lipoyl binding. This analysis doesn't account for the shared domain structure of E3BP with E2. Is E3BP functional in fungi? Does it have a different function than E2?

8.) Lines 283-288; Docking is performed in HADDOCK and only one pose is considered. Do other poses also satisfy possible linkages? Can you comment if they orient the domain in nonproductive orientations, or some other reason that they are not strongly considered outside of showing them in the SI?

9.) Line 297; should E81 be included with D79?

10.) Lines 335-336; "Here, we have established a single-step fractionation protocol to enrich for endogenous..." How is this different from the Cell Rep. paper?

11.) Line 345: the use of 'surprising' seems not needed. The ability of the LD to move between active sites is expected.

Methods:

12.) Line 393: "harvest at stage" appears to be incomplete. Regarding 'significant impact on observed PDHc activity' in lines 396, in what way? It is not clear.

13.) Line 396: "significant impact", please report statistical measures if using significant

14.) Line 401: 2 mM ThDP is included in the assay, although the main text indicated that the PDHc purified with this cofactor. Was loading of the cofactor checked, and excess included in the assay to deal with only partial retention?

15.) Immunoblotting experiments: Suggestion to report final concentrations rather than initial volumes

16.) Reference 54; update ID and DOI info.

Figures

17.) Figure 1 panel B; why is there a large shift in migration for PDHc E1beta?

18.) Figure 1 panel C; suggestion to include E3BP

19.) line 814, "active sides" should be "active sites"

20.) line 835, the use of 'clearly' should be considered at 3.8 Å resolution.

21.) Figure 2 panel A, what is the coloring scheme for E2 in the reconstructed map?

22.) Figure 2 panel C, how was coA positioned? If modeled, that should be clear in the caption.

23.) Figure 3 panel B, The beta strands are not visible

24.) line 878, indicate contour levels are for cryoEM maps

25.) Figure 5, it is not clear in the figure that ligands are from docking procedures

26.) Figure S10, It is difficult to see the surface representation and opening for the active site.

27.) Figure S11. It is surprising to see a docking model equilibrate in such little time with RMSD of less

than 1 Å and suggests that the structures really do not undergo much change. Is this because the overall protein orientations were restrained? If so, what is the result of unrestrained simulations on the protein-protein interaction over this timescale? Considering that a harmonic restraint is placed on the lipoyl and coA groups, the proteins should stay within an interaction sphere without needing to restrain the entire protein chains. The caption could be a little more descriptive as to the rationale for such a quick equilibration with minimal changes to the structure, and defining what "stability fo the system" means.

Reviewer #3:

Remarks to the Author:

In the present manuscript, the authors unravel the structural organization of the pyruvate dehydrogenase complex (PDHc) and provide molecular information about the transacetylase reaction at an unprecedented detail. To accomplish this goal, they combine Cryo-EM, biochemical assays, molecular docking and molecular dynamics. Briefly, from my point of view the results obtained are really interesting and the structural insights provided in this work will pave the way toward more detailed studies of the molecular mechanism underlying PDHc function. My comments will focus on the structural/modelling sections. In the title and abstract a lot of emphasis is put on obtaining mechanistic insights of the transacetylase reaction. Most of these observations rely on the predictions made from docking calculations and molecular dynamics simulations of the E2-LD assembly. I have a few concerns regarding the results obtained using these computational methods.

Major concerns:

1. The docking calculations performed using HADDOCK provide relevant information on the E2-LD binding mode. However some flexible parts of the E2 Nter domain are not included in the docking calculations (backfolded element for example). My question is whether the flexible Nter not observed in the core structure can play a role in stabilizing and properly orienting LD for catalysis once LD is bound? The authors should comment on the impact of not considering the flexible regions in the docking predictions and subsequent MD simulations.

2. Molecular Dynamics simulations are used to obtain structural insights into the transacetylase reaction. From my point of view the MD simulations performed are not extensive enough (based on current standards) to support the claims made in the last section. First, the simulation time is probably too short to evaluate the stability of the E2-LD complex. At least, 100 ns should be run to obtain reliable results. This simulation length is computationally accessible considering the size of the simulated complex. Second, only one replica of MD simulations is performed. The authors should perform at least three replicas of MD simulations to verify the results obtained. Third, in the methods section of the MD simulations the authors mention that they model ligand LA2. However, the molecular details of how LA2 ligand is modelled are not explained. In this particular case, the authors should specify how the covalent bond between Lys and lipoyl is treated in the MD simulations. Fourth, MD simulations are only analyzed in terms of RMSD. The authors should check the stability of the relevant interactions in the E2-LD interaction interface described in Figure 5. From MD simulations averaged distances can be obtained to confirm the stability of these interactions. Finally, it is not described how lipoylated Lys (from LD) is oriented with respect to CoA in the MD simulations. This interaction is key for the reaction carried out at E2 and analyzing the evolution of this interaction can provide additional information on the reaction mechanism. Overall, a more complete analysis of MD simulations should be performed and more information should be added in the supporting information and in the methods section.

Minor concerns:

3. To properly follow the reaction mechanism described in the introduction, it would be good to reference Figure 1 or to add a more detailed scheme in the supporting information combining

information of reaction cycles from Figures 1 and 5.

4. From my point of view it is difficult to understand where the intertrimeric regions are located. Maybe the authors can highlight these regions in Figure 1A or add an additional figure in the SI. In the Figure 2 caption the authors mention that the inter-trimeric interaction results from the dimeric interaction between two trimers, however, this is not mentioned in the main text (line 154). I think that it is relevant to clarify this aspect in the main text to understand how E2 trimers assemble.

5. Regarding the intra-trimeric arginine cluster, I wonder if there are other residues that contribute to the stabilization of this interaction. Does HADDOCK scoring functions used for the energetic analysis (Figure 2D) properly account for this kind of interaction that seems to play a key role in the intra-trimeric stabilization? In the sense that the quantum effects will be important to properly describe this interaction.

6. Figure 6A and 6B are not specified in the SI.

F. Feixas

Point-by-Point Response to the Reviewers' Comments

Please note that line numbers correspond to the generated PDF file of the main text from the submission system.

Reviewer #1:

Overview: In this manuscript Tuting et al study the enzyme complex pyruvate dehydrogenase (PDHc) from *C. thermophilum* lysate. This study is a continuation of the previous study (Kyrilis et al, Cell reports 2021), which put the molecular architecture of this complex forward. The advances of the study by Tuting et al compared to the previous study are:

- Improved resolution (E2p core from 6.9 Å to 3.85 Å, holocomplex: 14.2 Å to 6.21 Å)
- Biochemical analysis: Km values of pyruvate and CoA verified for prep
- Computational analysis of complex interface sizes and energetics
- Computational model of E3BP C-ter model fitted into core scaffold.
- Computational docking is used to derive a model for catalytic function of the enzyme complex, which is probed using molecular dynamics simulations.

Answer #1: We thank the Reviewer #1 for their short description of our results in our manuscript and our contributions to the advancement of understanding the giant PDHc metabolon.

Major comments:

Overall, the manuscript appears well written and the cryo-EM analysis is thorough within the resolution claims. On the technical side this reviewer notes that some interpretations, such as side chains and Pi-electron networks, typically base on resolutions beyond the 3.85 Å achieved in this study. Ultimately, this resolution appears to be limited by the magnification chosen for this study – which is arguably not an optimal choice for such a detailed structural dissection of the complex.

Answer #1.1: We thank the reviewer for their kind words and approving that resolution claims and interpretation is within resolution achieved for a native, metabolon-embedded PDHc scaffold. We have now moved claims of side-chains and Pi-electron networks in the supplementary material and added a paragraph in the discussion where we report our rationale behind chosen magnification. We would like to note to the reviewer that the resolution for the given Nyquist is 3.2 Å, and we reached 3.85 Å, meaning that there is still information to be harnessed at the applied magnification. We chose this magnification because the native PDHc 10 MDa complex is not very frequent in the fractions and it was a priority to increase number of particles used in the study; As the reviewer can appreciate, 3 PDHc molecules are on average identified per micrograph; By reducing the pixel size to 0.9612 Å (which we did e.g. for the apoferritin structure [PMID: 32374767], and accounting for the uselessness of the corner information in a cryo-EM micrograph, number of PDHc particles per image would be below 0.7 PDHc particles/image. Therefore, we believe that the chosen magnification for studying structural aspects of metabolons in complex mixtures is optimal for our cryo-EM set up and the increased complexity of the sample studied.

-We (a) explain the chosen pixel size in the Discussion section, (b) down-tone our results regarding side chain resolution and pi-electron networks and (c) removed Figure Panel 2B which showed high resolution features.

(a) Lines 365-374: Added section to the discussion, where the limitations of the pixel-size are being discussed.

(b) Lines 152-172: Rewrote the sentences concerning the resolution claims.

(c) Fig 2: We removed Panel B that exhibited high resolution features and down-toned the figure legend.

This reviewer is neither a specialist on docking/molecular dynamics nor on pyruvate dehydrogenase complexes. While the approach and the resulting swinging-arm model for enzyme function appear intriguing, this reviewer misses an effort to verify aspects of this model experimentally. For example, pyruvate, CoA or analogs, if available, could be used to shift the equilibrium of the reaction to obtain the 'CryoEM snapshots' of the reaction promised in the title.

Answer #1.2: We thank the reviewer for considering our work attractive and the resulting functional model for the transacetylase reaction intriguing. Using substrate or (non-hydrolysable) analogs to shift the equilibrium towards reaction intermediates and capture these using high-resolution methods is often applied. To get reproducible results, a very defined system is needed to fine-tune the substrate to enzyme concentrations. By using fractionated native cell extracts, we are facing a very high heterogeneity and relatively low PDHc concentrations. Additionally, other CoA-binding complexes like OGDHc and BCKDHc present in the sample influence the availability of added substrate. Therefore, these experiments should be done *in vitro*, using recombinant proteins, and not cell extracts. Indeed, such integration is useful and we discuss its implications in the revised manuscript. In addition, during our revision process, a manuscript on the bacterial, octahedral PDHc was released in Nat. Comm., complementing the recognition model that we propose for the eukaryotic complex. In addition, cryoEM snapshots refer to the imaging of the lysate, not to the reaction. We address the comment of the reviewer, in-text, below:

Lines 410-431: We explain the contribution of the helix Q227-N251 in the recognition and stabilization of the LD domain in combination with the recently released bacterial counterpart cryoEM map.

Lines 453-462: We added a paragraph in the discussion section, describing potential future biochemical experiments with reconstituted components to further understand the PDHc mechanism.

In summary, this manuscript constitutes an advance to the metabolon field. The combination of different experimental and computational methods and the developed functional model of the PDHc are appealing. However, this reviewer would suggest down-toning some of the mechanistic interpretation, which ultimately bases on rather low-resolution cryo-EM data and computational models that await further experimental validation.

Answer #1.3: We appreciate the reviewer's insights and we, once again, thank this reviewer for considering our functional model appealing. We have down-toned some of the mechanistic interpretation throughout the manuscript and included additional insights into the Discussion section on how the combination of cryo-EM and computational structural biology can provide unprecedented insights into pyruvate oxidation. We also pinpoint future work that can be done to further dissect the extremely complex function of the 10 MDa metabolon from native cell extracts.

Lines 152-167: Rewrote the sentences concerning the resolution claims

Lines 177-179: Down-toned the claims on the pi-electron network

Lines 185-187: Slightly re-wrote the sentence reporting on the pi-electron network

Lines 375-382: Added a paragraph to explain the combined experimental/computational modeling that eventually led to the analyzed molecular model.

Lines 383-388: Added a paragraph to rationalize the pi-electron network observed by our integrative approach.

Figure 2: The visibility of side chains is not compelling – which is not surprising at a resolution of ~4 Å.

Answer #1.4: We have now removed Figure 2B panel and down-toned the description in the main text and figure legend. Please see **Answers #1.1** and **#1.3** for further edits regarding resolution claims.

Fig 2: Panel B is now removed

Figure 2C/D: CoA is not resolved in the cryo-EM structure and the positioning of side chains is speculative at this resolution. Taken together, the panels may be misleading.

Answer #1.5: We clarified the origin of the CoA molecule, derived from the MD-refinement (see **Fig. 5**), in the figure description. The binding site of CoA in the active site is generally known (see PDB ID 1EAD), and our results are in agreement with this. We also agree with the reviewer, that at this resolution we cannot pinpoint a single rotamer of each of the side-chains, but for the majority of side-chains, we see at least density for C β -atom, which highly confines the conformational space.

Fig 2: We added the following text in the Figure legend, explicitly stating that the CoA is modelled “CoA molecule is not resolved but is here computationally placed and refined within the binding site as described in the Methods section”.

Pi-electron network remains a hypothesis at this level of resolution (Figure 2E), also in L. 167.

Answer #1.6: We have now stated clearly in-text that pi-electron network is a hypothesis and rationalize in our revised manuscript why; Although for these exact residues, as shown in Fig 2D, we have side-chain resolution and the Args fit in their densities, it could be that such a network might be a minor population due to the fact that cryo-EM maps come from averaged data, and therefore, at the single-particle level, Arg residues can acquire other rotamers, facing away from the pi-network. Still, our large-scale conservation analysis (Fig. S4) corroborates the critical importance of these Arg residues in fungi. Following the suggestion of the reviewer, we have down-toned the observation in-text and in the figure legend and added a section in the Discussion on how an Arg cluster can be further understood by e.g., applying quantum mechanical calculations.

Lines 177-179: Down-toned the identification of the π e-network.

Line 230: Added the word “probable” in “probable local delocalized π -electron networks”

Fig 2D: We removed the “ π e-network” annotation from the main figure (previously Fig 2E). We also added in the Figure legend “potential Arginine cluster”.

Lines 383-388: We added a short comment on possible averaging effects that could influence the observed proximity of the Arg chains.

While the resolution claim is 3.85 Å in the abstract, in figure 2 3.8 Å is mentioned. Please clarify.

Answer #1.7: We corrected this typo.

L. 151: how did the authors measure improvement of side-chain localization? Clashes? Ramachandran outliers?

Answer #1.8: Improvement is measured by the fact that the previous map we reported [PMID: 33567276] was resolved at 6.9 Å, whereas this reported map is resolved at 3.85 Å; therefore, better cross-correlation to the map densities is derived, which constitutes a drastic improvement over the previously reported side-chain fitted/refined model of the PDHc core. We clarified this now in this specific line.

Lines 152-159: Clarified the main text by reporting the cross-correlation between the map and the models using Phenix.

Lines 173-175: We also optimized our reporting of cross-correlation values

Lines 562-564: Updated the Methods where the Phenix cross-correlation values are described.

L. 151/152: how sensitive is definition of interfaces on modeling errors that will still be present for a model based on a ~ 4 Å EM structure?

Answer #1.9: Modelling errors in the 3.85 Å EM structure should be localized at the side chain rotamer level for residues of which the side-chains are not resolved. HADDOCK is a modelling software which optimizes the side-chain positioning considering a physics-based force field and calculates optimal energetics for interface residues. Therefore, refined interfaces with HADDOCK are well-defined as reported from the interface energetics and buried surface area statistics which compare very well with other natural interfaces found in protein-protein interactions [PMID: 24768922, 16043700]. To further consider the reviewers' comment we have added a comment in the Discussion section regarding quality of the models that we derived.

Lines 375-382: We added a short paragraph in the discussion, mentioning the missing side-chain densities and implications on the unambiguous placements of side-chains and the application of integrative methods like MD-simulations, to improve model quality.

Figure 5: How do the authors validate the (Haddock) model?

Answer #1.10: We validated the Haddock model utilizing Biochemical, Biophysical and Statistical observations. These include:

- (a) The Lys75 which gets lipoylated must face the binding site. The only solution that derived this kind of stereochemistry was Cluster 1, which we describe in the manuscript. In addition, when the lipoyl moiety is modelled, it must not clash with the binding site, and this was the case with Cluster 1 solutions.
- (b) The docked model includes the ordered domains of the LD and the core. Any unstructured regions were not used in modeling. When these regions are explicitly modelled, a correct solution should be clash-free. This is exactly what we observed when the unstructured regions for both domains were explicitly modelled (see RFig 1, where we verify the accessibility of this region by generating 5 *de novo* models of the docked solution, where we add the disordered region).

Figure R1. *De novo* modelling of flexible linkers. (A) Five independent docking solutions of the flexible linkers of the three E2 monomers in the selected docking solution (Cluster 1), generated by MODELLER. (B) Zoom-in into the LD domain. There is no clash between the linkers and the LD domain, indicating sufficient conformational space for their accommodation.

Lines 306-317: We reorganized the result part and added a summarizing sentence.

Fig S11: We added a new plot showing the clashes between the anchor points. We only display now Clusters including more than 15 clustered models from a total of 500 water-refined models.

L. 164: cross-correlation coefficients depend on parameters such as the low pass filter used. It may be more informative to compare the 60-mer model with decoys (or the previously published model).

Answer #1.11: We thank the reviewer for pointing this out. We now compare the entire 60-mer with our previous published model, using the PHENIX “Comprehensive validation (cryo-EM)” tool, which reports cross-correlation coefficients. We added the respective data into the manuscript.

Lines 152-159: Clarified the main text by reporting the cross-correlation between the map and the models using Phenix.

Lines 562-564: Updated the Methods where the Phenix cross-correlation values are described.

Minor comments:

- It should probably read “... the PDHc core’s unusual stability.”
- Minor: the text has relatively few paragraphs, which makes it a bit difficult to follow the text. Limitation to one thought per paragraph would help.
- Minor: the figure captions include description of results that this referee would have expected in the main text.

Answer #1.12: All minor comments have been edited according to reviewer’s instructions. However, we decided to keep Figure captions as they are, because this coarse description of presented results in the figure panels can aid the non-expert reader to better appreciate the presented content.

Reviewer #2

Overview: The article entitle “CryoEM snapshots of a native lysate provide structural insights into a metabolon-embedded transacetylase reaction” by Christian Tuting, Fotis Kyrilis, et al. reports the 3.85 Å resolution cryoEM map for the E2 icosahedral core of the pyruvate dehydrogenase complex (PDHc) from the thermophilic filamentous fungus *Chaetomium thermophilum*. A map of 6.21 Å resolution when considering only C2 symmetry revealed map density for part of the E3BP subunits within the core, arranged in a tetrahedron of trimers. The generated model was analyzed to reveal only ~25% of the core is filled, mostly by the E3BP minimal fold and inner loops of the E2. Docking of the Lipoyl domain (LD) onto the generated protein model identified possible binding modes that would position the lipoyl group into the active site adjacent to the coA binding site. The authors report that their preparation of endogenous PDHc metabolon in a heterogeneous, high-molecular weight cell extract fraction was key to obtaining enough particles for map construction of homogenous species. Overall, the biological significance of the work is clear in the difficulty of identifying structures of PDHc, a very complex complex with multiple subunits that links glycolysis and the TCA cycle in central metabolism. The obtained structure of the PDHc core at sub 4 Å is an improvement to previous reported maps for fungi PDHc cores, as the current maps show better definition of helices and some density for side chains. The method for sample preparation was recently published by the author’s in *Cell Rep.*, leaving the extension of map resolution and the ability to do additional analysis based on those results to be the major findings and reporting of this work. The model is overall similar to the lower resolution models in *Cell Rep.* and the complex is consistent with work by Forsberg et al. *Nat. Comm.*, but the increase in resolution to 3.85Å is notable and therefore worthy of considering for publication. The analysis is nicely performed and the article is well written. A few queries for the authors are below.

Answer #2: We thank the Reviewer #2 for appreciating the impact and results of our manuscript and noting that our concept can provide high-resolution information for such large and heterogeneous complexes directly from endogenous cell extracts.

Major comments:

The FSC (Figure S1) shows a dip around 0.18 1/Å for both the masked and unmasked maps. What is the source of this dip?

Answer #2.1: The source of this dip is the presence of internal and external densities of the PDHc core, corresponding to the E3BP and the E1 and E3 proteins. E3BP has tetrahedral symmetry, while E1 and E3 proteins are asymmetrically distributed around the icosahedral core scaffold, an intriguing observation which we reported previously [PMID: 33567276]. As the reviewer can appreciate from the rest of the FSC plots, this dip disappears by applying lower symmetry, revealing the positioning of E3BP inside the core structure. We have now added this comment at the legend of Fig. S3 (previous Fig. S1).

Fig S3: Edited figure and legend to explain the origin for the observed FSC dip.

1.) Results section 1, lines 103-129; The details that enabled obtaining a good sample of PDHc are not clear in this section, and the methods section doesn't add much more detail. "Note that the growth state of *C. thermophilum* before harvesting significantly impacted the retrieved PDHc activity and was properly adjusted to recover optimal PDHc activity" Adjusted in what way?

Answer #2.2: We thank the reviewer for their comment. During initial cultivation from solid culture on plates to liquid culture, different sizes of mycelia are visible. At a certain size of the mycelia, the inside cells are most likely in cell arrest or already dead and only the cells on the surface are viable. To optimize our PDHc yields, we passaged the liquid culture several times by taking the turbid media (containing new, very small mycelia). By this, we obtain cells at closer cell cycle stages. We were able to retrieve many small mycelia of the same size, instead of a few bigger mycelia, heterogeneous in size (like in the starting culture). We regularly verify cell lysates for PDHc/OGDHc activity in quick screening experiments and we see a trend of higher activity in cultures with smaller, more uniform mycelia. We rephrased this section and removed "significant".

Lines 470-481: We updated the method section on how we optimize the cultivation conditions.

2.) Line 125; report error for Km values

Answer #2.3: We now have added the standard deviations for reported Km values.

Line 131: Updated Km values.

3.) lines 179-181; It is unclear what/where loop is positioned relative to the core. Line 175 states 'inward facing loop', but this is unclear. Inward towards what?

Answer #2.4: We replaced "inward facing loop element" by "loop element facing inside the core cavity" to clarify, that we are talking about the loop element of the E2 monomer, which is inside the icosahedral core structure.

Lines 188-195: Slightly updated main text as described above

4.) Line 203; two antiparallel beta-strands would make a beta-beta motif. Are there more arranged together to warrant description of a sheet? Is this a hairpin? The beta-strands are not visible in the figure.

Answer #2.5: We appreciate the reviewer's comment. We have now added an additional panel in Fig. 3 to show the 2 antiparallel beta-strands of E3BP and edited their description in the Figure panel, legend and in the main text as "beta-beta motif". Because the density that

we captured is only the rigid part of the sequence, it is unknown if a beta sheet is formed. Therefore, we describe this identified element, as the reviewer suggest, as a beta-beta motif.

Fig 3: Updated figure panel B to show the two beta strands.

Lines 218-220: Changed beta strands to beta-beta motif

5.) Line 207; the use of 'remarkably' seems unjustified based on the sentence. Rather it is consistent or expected.

Answer #2.6: We replaced the word “remarkably” with “As expected,” (Line 222).

6.) Lines 227-228; regarding "Phe330 from E3BP in a hydrophobic pocket in the E2 core". This implies hydrophobic/VDW forces would be contributing, but this is counter to the previous sentence.

Answer #2.7: We thank the reviewer for the comment. Desolvation energy in HADDOCK is connected to the hydrophobic effect [PMID: 14687579]. Here, the interface is very small due to its transient nature, and, therefore, a major contributor for desolvation is the hydrophobic side chain of the Phe330 residue.

Lines 243-245: We simplified the sentence to discuss the hydrophobic contribution and added the corresponding reference.

7.) Section lines 237-267: A link is made between proximity of E2 and E3BPs being in an 'open' state, whereas E2 at the periphery of the core, and away from E3BPs, are more likely in a closed state with a 'back-folded element' positioned over the active site and preventing lipoyl binding. This analysis doesn't account for the shared domain structure of E3BP with E2. Is E3BP functional in fungi? Does it have a different function than E2?

Answer #2.8: We thank the reviewer for their useful comment. We discuss the localization of E3BP in our manuscript: We presented its role in the introduction, where we cite relevant publications regarding its localization. Briefly, for fungal PDHc, like yeast and also *N. crassa* and *Chaetomium*, E3BP is a distinct protein inside the core of the homo-60meric E2 complex. It was shown that E3BP could bind after the core is formed. E3BP is not functional in terms of the transacetylase reaction and CoA binding, and we added this information to the introduction. In mammals, E3BP is part of the core forming unit. The stoichiometry of either 40:20 or 48:12 E2 to E3BP for mammalian PDHc is cited. Therefore, our analysis in this section indeed accounts for the E3BP, because it is only present inside the PDHc core in fungi.

Lines 49-67: We also slightly edited the paragraph to make the content clearer, added a sentence in the introduction about the lack of catalytic activity of E3BP and added the Fig. S1 for further understanding of the complex underlying reaction.

8.) Lines 283-288; Docking is performed in HADDOCK and only one pose is considered. Do other poses also satisfy possible linkages? Can you comment if they orient the domain in nonproductive orientations, or some other reason that they are not strongly considered outside of showing them in the SI?

Answer #2.9: The docking was just restrained by the Euclidean distance between the C α of the lipoyllysine in the LD and the anchor point in the active site in the E2 core. Please see also **Answer #1.10**, where a similar remark was answered. In short, our selection of only 1 cluster relies on the additional knowledge: (a) localization of the lipoyllysine in respect to the binding pocket and accessibility of its moiety in the active site unhindered, and (b) the occupancy of the N-ter of E2, where the flexible linkers are located and are not clashing when explicitly modelled. We updated the text and also the supplementary figure to address the reviewer's concern. Of course, we cannot exclude that any of the structures within the cluster could be optimal, but considering the transient nature of the complex, it is highly likely that an encounter, “fuzzy” complex is formed.

Lines 298-317: Updated main-text to include the biochemical validation for the selection of the displayed solution.

Fig S11: Additional panel B was added to show that all solutions from Cluster 1 are having in Lys75 directly accessible to the binding site. We showed the most energetically favoured pose of Cluster 1 in the manuscript and for subsequent MD analysis.

9.) Line 297; should E81 be included with D79?

Answer #2.10: Yes, we changed the text according to this suggestion (**Lines 322-323**).

10.) Lines 335-336; "Here, we have established a single-step fractionation protocol to enrich for endogenous..." How is this different from the Cell Rep. paper?

Answer #2.11: The reviewer is right, there are minor differences that are not worth mentioning in the main text and are described in the Methods (e.g., optimized cultivation conditions, bead beating protocol). We have rephrased the sentence (**Lines 359-360**) as follows: "Here, we have optimized our previously established single-step fractionation protocol [PMID: 33567276] to enrich for endogenous..."

Lines 470-481: We updated the method section on how we optimize the cultivation conditions.

11.) Line 345: the use of 'surprising' seems not needed. The ability of the LD to move between active sites is expected.

Answer #2.12: We removed the word "surprising" and now reads as: "pointing to an intricate role of localized flexibility". (**Line 393**).

12.) Line 393: "harvest at stage" appears to be incomplete. Regarding 'significant impact on observed PDHc activity' in lines 396, in what way? It is not clear.

Answer #2.13: We rewrote the methods part, highlighting the differences to our previous publication. We also removed the word "significant", as our observation of higher PDHc yield is the result of activity analysis and electron microscopy screening carried out as standard after cell lysis and fractionation. As a rule of thumb, if the mycelia in the liquid culture are around 1 cm in diameter and equal in size, we can expect an optimal yield for protein complexes of the aerobic metabolism, including PDHc and OGDHc, which is expected from cells in the log phase of growing.

Lines 470-481: We updated the Methods section on how we optimize the cultivation conditions.

In addition, we performed slight modifications in the whole Methods section with the aim to be more thorough and detailed for the reader.

13.) Line 396: "significant impact", please report statistical measures if using significant

Answer #2.14: We removed significant. See also **Answer #2.13**.

14.) Line 401: 2 mM ThDP is included in the assay, although the main text indicated that the PDHc purified with this cofactor. Was loading of the cofactor checked, and excess included in the assay to deal with only partial retention?

Answer #2.15: ThDP (or TPP) is bound only non-covalently to the E1 subunit. During purification, this cofactor might be partially lost. Due to the heterogeneity of the sample, we did not check the catalytic state of the PDHc E1 enzymes. Also, titrating TPP to calculate free E1 is not suitable, as also OGDHc and BCKDHc metabolons are present in this sample, and both also have a E1 subunit which binds TPP in the active site. Therefore, to saturate all E1 subunits with TPP, we added 2 mM to the reactions, to not limit our reaction and thereby retrieve wrong results. During revision, we recognized that we confuse the two references in

this Methods section, which we corrected. Reference 47 now correctly describes the assay in detail, including addition of 2 mM TPP to the reaction.

15.) Immunoblotting experiments: Suggestion to report final concentrations rather than initial volumes

Answer #2.16: We now additionally report exact amounts of protein loaded. To harmonize the writing, we also exchange all volumetric values to concentrations (e.g., acrylamide in gel). In addition, immunoblots are now shown uncropped in new **Figure S2**.

Lines 493-502: Revised the Methods section according to reviewer's suggestion and slightly re-wrote **Lines 517-518**.

16.) Reference 54; update ID and DOI info.

Answer #2.17: We updated this reference according the *Nature* citation style for datasets, now it is reference 62.

17.) Figure 1 panel B; why is there a large shift in migration for PDHc E1beta?

Answer #2.18: The positive control is overexpressed in *E. coli* and thereby lacking any post-translational modifications. On the other hand, the endogenous protein might be modified, e.g., by phosphorylation which increases the negative charges during SDS-PAGE separation, resulting in a lower apparent molecular weight. We can also not rule out, that due to signal-peptide cleavage on the endogenous protein, we have a significantly shorter polypeptide than the one used for overexpression, which also carries an additional His₆-tag. We added a comment regarding this shift on the uncropped western-blot images in the supplementary information.

Fig S2: It is now first mentioned in **Line 120** and is added an additional supplementary figure, showing the uncropped western-blots. The figure legend now describes possible reasons for the apparent shift.

18.) Figure 1 panel C; suggestion to include E3BP

Answer #2.19: This figure panel illustrates the active sites required for the pyruvate oxidation. E3BP does not have an active site for catalysis, but plays a role in binding E3. We leave the Figure 1C therefore as it is. Nevertheless, to include also E3BP in the reaction schema, we included a new Supplementary Figure (Fig. S1), which shows the role of E3BP in the PDHc metabolon.

Fig S1: Detailed schema of the fungal PDHc metabolon and its catalyzed reactions. It is now mentioned for clarity in **Lines 55** and **292-293**.

19.) line 814, "active sides" should be "active sites"

Answer #2.20: We thank the reviewer for pointing out this typo, we updated the term in the revised version, both in the main manuscript and in the supplementary.

20.) line 835, the use of 'clearly' should be considered at 3.8 Å resolution.

Answer #2.21: According the suggestions from Reviewer 1 (please see **Answers #1.1** and **#1.3**), we removed this subpanel and generally down-toned resolution claims throughout the manuscript.

21.) Figure 2 panel A, what is the coloring scheme for E2 in the reconstructed map?

Answer #2.22: We actually just use the default rainbow palette coloring of ChimeraX that color-codes each E2 monomer uniquely. We clarified this in the figure description.

Fig 2: Updated figure legend to describe the chosen color code.

22.) Figure 2 panel C, how was coA positioned? If modeled, that should be clear in the caption.

Answer #2.23: The CoA molecule was superimposed from the MD refined E2-core/LD complex (see Fig. 5 and Methods for details). We clarify this now in the figure legend.

Fig 2: Updated figure legend.

23.) Figure 3 panel B, The beta strands are not visible

Answer #2.24: We have added an additional cartoon representation (red cartoon model) showing the overall structure of E3BP that includes the beta-beta motif.

Fig 3: Updated Figure panel B

24.) line 878, indicate contour levels are for cryoEM maps

Answer #2.25: We have described in the figure legend that those contour levels are for the cryoEM maps.

Fig 2 & Fig3: Updated Figure Legends as per reviewer's instructions

25.) Figure 5, it is not clear in the figure that ligands are from docking procedures

Answer #2.26: We updated the figure description, clearly stated that the lipoate and CoA are derived from the MD simulation.

Fig 5: Updated figure legend, writing: "The lipoate and CoA were derived from docking procedures and refined during MD simulation (see Methods)."

26.) Figure S10, It is difficult to see the surface representation and opening for the active site.

Answer #2.27: We re-organized the Figure (now **Fig. S10**) to illustrate the fact that the human and *N. crassa* LD-core models have inaccessible sites, whereas the *C. thermophilum* counterpart could bind the LD without clashes. This is shown now in (A-C) panels using the orange line (which shows the space where the lipoate is bound), which is more contrasty in the *C. thermophilum* depicted model. To further show the difference to the reviewer statistically, we added a new analysis shown in panel D. This panel shows that Cluster 1 of *C. thermophilum* can accommodate the lipoyl because the number of clashing atoms within this distance between the Lys75 and the CoA is very small; This is in contrast to the measured distances in the generated models for Clusters 1-3 for human and *N. crassa*.

Fig. S12: Reorganized the Figure and added a new panel, showing the number of atom clashes in the lipoyl binding pocket.

27.) Figure S11. It is surprising to see a docking model equilibrate in such little time with RMSD of less than 1 Å and suggests that the structures really do not undergo much change. Is this because the overall protein orientations were restrained? If so, what is the result of unrestrained simulations on the protein-protein interaction over this timescale? Considering that a harmonic restrain is placed on the lipoyl and coA groups, the proteins should stay within an interaction sphere without needing to restrain the entire protein chains. The caption could be a little more descriptive as to the rationale for such a quick equilibration with minimal changes to the structure, and defining what "stability of the system" means.

Answer #2.28: We previously performed restrained MD simulation to (a) further relax the docking model from HADDOCK after explicit solvent refinement and (b) to optimize the lipoate and CoA localization which were not included during docking. Previous MD were performed over few ns. Considering this comment of Reviewer #2 and the comments from Reviewer #3, we have now:

(a) re-performed the "hard-restrained" MD in triplicate to improve statistics

(b) expanded each simulation to 100 ns to show system equilibration in a longer time scale

- (c) performed additional MD (“soft-restrained”) using only a harmonic restraint between the lipoyl and CoA groups, again in triplicate over 100 ns each
- (d) performed additional analysis concerning the newly performed MDs and the resulting interactions between
 - i. lipoyl and CoA, discovering critical flexibility for the adenosine moiety observed in both types of simulations
 - ii. LD and E2 core, validating
 - the stability of the positioning of the LD interacting mostly with 1 E2 core monomer
 - the critical charge-charge interaction (the key interaction Arg307 x Asp79/Glu81) discovered during the docking exercise.

The reviewer can visualize the derived movies, currently available via the following link:

<https://cloud.uni-halle.de/s/oh71bEvAvZRvI7c>

All files related to our manuscript, including the MD, are available via the SBGrid deposition for reproducibility and transparency.

Our results are shown in **Figure S13** and explained at the Figure Legend:

“Figure S13. Detailed analysis of the performed MD-Simulations. (A) “Hard-restrained MD” triplicate across 100 ns – RMSD plot. The plot shows overall stability for each MD simulation. (B) LA2 from initial frame was used as a reference to calculate mean distance of its atoms across the simulation. The plot shows fast relaxation and stable localization over time. (C) Same as (B), but for CoA. In one of the three simulations, after 60 ns, flexibility is observed. (D) Explanation of the observed flexibility in (C); The CoA adenosine moiety undergoes conformational rearrangements, escaping the charge complementarity imposed by its binding site. This observation possibly shows an initial mechanism for its release. (E) “Soft-restrained” MD triplicate across 100 ns – RMSD plot. The plot shows increased flexibility due to the applied protocol while 2/3 simulations show relative stability. MD5, after ~75 ns shows increased flexibility due to loop movements in the unbound E2 chain. (F) Relative distance of the calculated center of mass (COM) between the LD and the bound E2 normalized to the initial frame calculated from the “soft-restrained” MD simulations. The COMs remain in relative proximity across the replicates. (G-H) Distance between the electropositive and electronegative atoms of Arg307 and Asp79/Glu81 present on the bound E2 core and the LD, respectively. Distances were calculated from the “soft restrained” MD simulations. These residues form an ionic interaction predicted by docking; The MD simulations show that the ionic interaction reoccurs across the simulation and across replicates; More frequent ionic interactions are observed between Glu81 and Arg307 due to the larger flexibility and longer side-chain of the glutamate residue.”

Overall, we have performed the above-mentioned MD, share the data via SBGrid, and include the protocols that we applied in the Methods section.

Fig. S13: We replaced the previous Figure with **Fig S13** including new results from the MD.

Lines 320-323: Slightly changed the corresponding text.

Lines 621-653: We describe the newly performed MD in the Methods section in detail.

Reviewer #3

Overview: In the present manuscript, the authors unravel the structural organization of the pyruvate dehydrogenase complex (PDHc) and provide molecular information about the transacetylase reaction at an unprecedented detail. To accomplish this goal, they combine Cryo-EM, biochemical assays, molecular docking and molecular dynamics. Briefly, from my

point of view the results obtained are really interesting and the structural insights provided in this work will pave the way toward more detailed studies of the molecular mechanism underlying PDHc function. My comments will focus on the structural/modelling sections. In the title and abstract a lot of emphasis is put on obtaining mechanistic insights of the transacetylase reaction. Most of these observations rely on the predictions made from docking calculations and molecular dynamics simulations of the E2-LD assembly. I have a few concerns regarding the results obtained using these computational methods.

Answer #3: We thank the reviewer for appreciating the insights and importance of our work. Indeed, we combined different biophysical, biochemical and biocomputational methods to probe the structure of the 10-MDa endogenous PDHc metabolon. We would like to point out that the scope of our work was not to perform extensive MD for the E2-LD assembly, but rather to relax the docking solution from HADDOCK and look into the derived interface from the docking calculation. However, we have followed the reviewer's suggestion to expand on the MD part, please see the answer to their comment below as well as answer to Reviewer #2, **Answer #2.28**.

Major comments:

1. The docking calculations performed using HADDOCK provide relevant information on the E2-LD binding mode. However some flexible parts of the E2 Nter domain are not included in the docking calculations (backfolded element for example). My question is whether the flexible Nter not observed in the core structure can play a role in stabilizing and properly orienting LD for catalysis once LD is bound? The authors should comment on the impact of not considering the flexible regions in the docking predictions and subsequent MD simulations.

Answer #3.1: We thank the reviewer for their useful comment.

The flexible parts of the E2 *N-ter* domain were not observed in the cryoEM structure, indicating their plasticity, and therefore, were not modelled. We showed in Fig. 3 that not only the flexible parts of E2, but also those of E3BP are close to the E2 core and, possibly the binding site. In addition, we show in Fig. 4 that positioning of the E3BP inside the core correlates with absence of extended *N-ter* densities, indicating that indeed the accurate E3BP positioning inside the core scaffold correlates with an open active site. Therefore, we agree with the reviewer that the flexible E2 *N-ter* part could have function in orienting and stabilizing the LD. However, including this region in docking calculations is not optimal in our opinion, because its structure is currently completely unknown, highly flexible and has very high degrees of freedom that cannot be reasonably sampled by any docking/MD simulation software. Therefore, we implicitly accounted for its presence, showing that the best docking cluster solution does not overlap/clash with the space that this flexible region should sample (please see Fig. R1 in **Answer #1.10**). In addition, in the *E. coli* structure that was released during the revisions of our manuscript, the LD is bound on the bacterial octahedral core with the N-ter in a partially folded conformation, and its comparison with our docking and MD simulations is included in the Discussion.

We have addressed this comment in the manuscript under sections Results and Discussion. In detail:

Lines 314-316: We added an explicit statement on the possible impact of disordered parts in docking calculations.

Lines 410-417: We additionally discuss the influence of not explicitly considering disordered regions on the predicted complex.

Lines 418-431: We additionally discuss the comparison to the *E. coli* E2 cubic core released during revisions of our manuscript and citing the reference.

2. Molecular Dynamics simulations are used to obtain structural insights into the transacetylase reaction. From my point of view the MD simulations performed are not extensive enough (based on current standards) to support the claims made in the last section. First, the simulation time is probably too short to evaluate the stability of the E2-LD complex. At least, 100 ns should be run to obtain reliable results. This simulation length is computationally accessible considering the size of the simulated complex. Second, only one replica of MD simulations is performed. The authors should perform at least three replicas of MD simulations to verify the results obtained. Third, in the methods section of the MD simulations the authors mention that they model ligand LA2. However, the molecular details of how LA2 ligand is modelled are not explained. In this particular case, the authors should specify how the covalent bond between Lys and lipoyl is treated in the MD simulations. Fourth, MD simulations are only analyzed in terms of RMSD. The authors should check the stability of the relevant interactions in the E2-LD interaction interface described in Figure 5. From MD simulations averaged distances can be obtained to confirm the stability of these interactions. Finally, it is not described how lipoylated Lys (from LD) is oriented with respect to CoA in the MD simulations. This interaction is key for the reaction carried out at E2 and analyzing the evolution of this interaction can provide additional information on the reaction mechanism. Overall, a more complete analysis of MD simulations should be performed and more information should be added in the supporting information and in the methods section.

Answer #3.2: We are very grateful to the Reviewer and their valuable insights. The MD simulation was performed to relax the active site after docking with HADDOCK and manual placement of CoA (slightly modified from PDB-ID 1EAD) and the lipoyllysine (placed with COOT). We had previously omitted details in the methods section and have now added a new paragraph, describing how we built these. The conformation and localization of both were not optimal, and also the side-chains of the residues in the active site were not in a “bound” state. We thereby constrained the distance between the SH-group of the LA2 and the SH-group of CoA to 3.5 Å and only allow atoms in a radius of 8 Å around these two cofactors to freely move.

We also agree with the reviewer that the initial MD simulation does not fulfil today’s quality requirements, even though it was used to relax the interaction derived from docking. To answer the reviewer’s comment, we now extended the MD simulation to 100 ns and performed the analysis in triplicates, performing in total 6 MD simulations of 100 ns each. We also updated our method section according to the reviewer’s comments. For analyzing the derived results, we now not only plot the RMSD as a function of simulation time, but we also plotted movements of LA2 and CoA, visualized flexibility of the CoA and calculated average distances of key residues (Arg307 with Asp79 and Glu81) that stabilize the substrate/cofactor in the active site. Please see also complementary **Answer #2.28** for further insights into the revised MD results.

Lines 612-619: Added a part in the Methods section for building the modified model

Lines 621-653: Added a part in the Methods section describing the extension of the MD simulation

Fig. S13: We replaced this Figure with the new results from the MD simulations and re-wrote completely the Figure legend.

Minor comments:

3. To properly follow the reaction mechanism described in the introduction, it would be good to reference Figure 1 or to add a more detailed scheme in the supporting information combining information of reaction cycles from Figures 1 and 5.

Answer #3.3. A detailed reaction pathway was added as supporting information and referenced in the introduction.

Fig S1: Detailed scheme of the reactions at each active site of the PDHc added.

4. From my point of view it is difficult to understand where the intertrimeric regions are located. Maybe the authors can highlight these regions in Figure 1A or add an additional figure in the SI. In the Figure 2 caption the authors mention that the inter-trimeric interaction results from the dimeric interaction between two trimers, however, this is not mentioned in the main text (line 154). I think that it is relevant to clarify this aspect in the main text to understand how E2 trimers assemble.

Answer #3.4: The reviewer is completely right; we did not clarify in this publication the interaction interfaces in the core structure, which make the results difficult to interpret if one is not familiar with the architecture of this complex.

Fig 2: We added a panel, where we indicate the two different interfaces in the core structure.

5. Regarding the intra-trimeric arginine cluster, I wonder if there are other residues that contribute to the stabilization of this interaction. Does HADDOCK scoring functions used for the energetic analysis (Figure 2D) properly account for this kind of interaction that seems to play a key role in the intra-trimeric stabilization? In the sense that the quantum effects will be important to properly describe this interaction.

Answer #3.5: This is an intriguing suggestion from the reviewer; Yes, HADDOCK accounts for the Arg cluster in the energetic analysis, and is in agreement with the showed increased electrostatics. According to **Answer #1.6** and the comments of Reviewer #1, we down-toned our results. Due to symmetrization, we cannot unambiguously identify this arginine cluster. In close proximity are the residues Thr257, Glu383 and Thr427 of each monomer, which could interact with the Arg384, forming H-bonding networks or ionic interactions. We added a short paragraph in the discussion section regarding this.

Line 383-388: Paragraph added to the discussion

6. Figure 6A and 6B are not specified in the SI.

Answer #3.6: We apologize for the typo, the SI panels were now labeled to properly show what is cited in the text.

Fig S8: Updated panels (previous Fig. S6).

Reviewers' Comments:

Reviewer #1:

Remarks to the Author:

The authors have addressed the points raised by reviewer #1 thoroughly. The more extended simulations indeed potentially strengthen the functional implications of the work. Reviewer #3 may be best suited to comment on the technical correctness.

This reviewer does not completely agree with answer #1.1, although this is a rather minor aspect. The appeal of the approach presented here is that substantial insights into structure are obtained without investing into reconstitution and protein chemistry. The presented approach to structurally characterize protein complexes using lysate is attractive, although ultimately at the expense of achievable resolution and mechanistic insight. Specifically, the reviewer does not quite agree with the conclusion that the magnification is not a major resolution-limiting factor (l.510). Differing from the authors' opinion, this reviewer considers ~85% of the Nyquist frequency the maximum that is realistically achievable with the equipment used by the authors (which is in fact a nice achievement). Thus, in this reviewer's opinion the magnification used is a major, if not THE major resolution-limiting factor.

Reviewer #2:

Remarks to the Author:

The authors of "CryoEM snapshots of a native lysate provide structural insights into a metabolon-embedded transacetylase reaction" have resolved my queries and in the process made improvements to areas of the manuscript that have added information. In particular, their MD methods have been fleshed out to report a level of detail for publication that also reveals added information. In rereading the manuscript, I wanted to mention three minor queries that arose.

1- Line 79 of the WORD file: The transfer from FADH₂ to NAD⁺ involves both electrons and protons.

2- Lines 134-137 of the WORD file: TPP is stated as "attached in their respective active sites, and therefore, always present." This is contradictory to the information provided by the authors in their letter for why additional TPP was used in the experiments "TPP is bound only non-covalently to the E1 subunit". Please verify and update the final manuscript accordingly.

3- Line 271 of the WORD file: in line with the decreased resolution, it is suggested to limit the report of the measured distance here to ~3.3 Å

Reviewer #3:

Remarks to the Author:

The authors significantly improved the manuscript following the suggestions pointed out by the reviewers. In particular, I find that the discussion of the results based on molecular docking and molecular dynamics simulations is now consistent and well explained in both main text and supplementary information. It is also interesting to see the significant conformational changes along the three replicates of 100 ns MD simulations; these results will require further evaluations in the future because relevant insights for the transacetylase reaction may be established. Overall, the work is interesting and will pave the way toward more detailed studies of the molecular mechanism (both experimental and computational) underlying PDHc function and dynamics.

Response to the Reviewers' comments (short).

We would like to deeply thank all 3 reviewers for their insightful comments, which we answer below. In addition, we have considered all editorial comments and are now addressed in the revised version of our manuscript according to the instructions from the Authors' checklist provided by the Editor. Short answers are given below to the Reviewers:

Reviewer #1 (Remarks to the Author):

The authors have addressed the points raised by reviewer #1 thoroughly. The more extended simulations indeed potentially strengthen the functional implications of the work. Reviewer #3 may be best suited to comment on the technical correctness.

Answer #R1.1. We thank the reviewer for their comment on our extended MD simulations.

This reviewer does not completely agree with answer #1.1, although this is a rather minor aspect. The appeal of the approach presented here is that substantial insights into structure are obtained without investing into reconstitution and protein chemistry. The presented approach to structurally characterize protein complexes using lysate is attractive, although ultimately at the expense of achievable resolution and mechanistic insight. Specifically, the reviewer does not quite agree with the conclusion that the magnification is not a major resolution-limiting factor (1.510). Differing from the authors' opinion, this reviewer considers ~85% of the Nyquist frequency the maximum that is realistically achievable with the equipment used by the authors (which is in fact a nice achievement). Thus, in this reviewer's opinion the magnification used is a major, if not THE major resolution-limiting factor.

Answer #R1.2. We thank the reviewer for their comment. We updated the manuscript text where we wrote: "Also, the chosen pixel size of 1.567 Å² is preferable for this kind of sample, even though this limits achievable resolution to 3.2 Å. Our reconstructed PDHc core at 3.85 Å indicates that chosen pixel size is not the main factor limiting resolution. Additionally, with higher magnification the average PDHc particle per micrograph would be reduced from ~3 to ~0.7, which disproportionately increases microscope acquisition time and downstream analysis.", into:

"Also, the chosen pixel size of 1.567 Å is preferable for this kind of sample. The magnification used is the major resolution limiting factor considering that ~85% of the Nyquist frequency is the maximum that is realistically achievable with our cryoEM equipment. However, with higher magnification the average PDHc particle per micrograph would be reduced from ~3 to ~0.7, which disproportionately increases microscope acquisition time and downstream analysis."

Reviewer #2 (Remarks to the Author):

The authors of "CryoEM snapshots of a native lysate provide structural insights into a metabolon-embedded transacetylase reaction" have resolved my queries and in the process made improvements to areas of the manuscript that have added information. In particular, their MD methods have been fleshed out to report a level of detail for publication that also reveals added information. In rereading the manuscript, I wanted to mention three minor queries that arose.

Answer #R2.1. We thank the reviewer for this comment.

1- Line 79 of the WORD file: The transfer from FADH₂ to NAD⁺ involves both electrons and protons.

Answer #R2.2. We changed the sentence from “Eventually, the reduced FAD is recovered by transferring the protons onto NAD+.”, to:
“Eventually, the reduced FADH₂ is recovered by transferring the two protons and two electrons onto a NAD⁺ molecule.”

2- Lines 134-137 of the WORD file: TPP is stated as “attached in their respective active sites, and therefore, always present.” This is contradictory to the information provided by the authors in their letter for why additional TPP was used in the experiments "TPP is bound only non-covalently to the E1 subunit". Please verify and update the final manuscript accordingly.

Answer #R2.3. This is a very good comment. TPP is bound in the active site of E1 and undergoes no association/dissociation during catalysis and is therefore considered “always present”. We add TPP to saturate the E1 active site in case any TPP is lost during purification. We updated the final manuscript accordingly by
a) removing “, and therefore, always present” in line 163
b) adding “the soluble substrates” in line 164, to highlight, that these undergo measurable turn-over during catalysis.

3- Line 271 of the WORD file: in line with the decreased resolution, it is suggested to limit the report of the measured distance here to ~3.3 Å

Answer #R2.4. We thank the reviewer for the comment. We prefer to keep the reporting of distances as they are in the text, since they are reported on atomic models.

Reviewer #3 (Remarks to the Author):

The authors significantly improved the manuscript following the suggestions pointed out by the reviewers. In particular, I find that the discussion of the results based on molecular docking and molecular dynamics simulations is now consistent and well explained in both main text and supplementary information. It is also interesting to see the significant conformational changes along the three replicates of 100 ns MD simulations; these results will require further evaluations in the future because relevant insights for the transacetylase reaction may be established. Overall, the work is interesting and will pave the way toward more detailed studies of the molecular mechanism (both experimental and computational) underlying PDHc function and dynamics.

Answer #R3.1. We are happy and appreciate that all comments of Reviewer #3 are satisfied.